# Loss of YhcB results in dysregulation of coordinated peptidoglycan, LPS and phospholipid synthesis during *Escherichia coli* cell growth

Emily C. A. Goodall[1]*, Georgia L. Isom[2], Jessica L. Rooke[1], Karthik Pullela[1], Christopher Icke[1], Zihao Yang[1], Gabriela Boelter[2], Alun Jones[1], Isabel Warner[1], Rochelle Da Costa[1], Bing Zhang[1], James Rae[1], Wee Boon Tan[3], Matthias Winkle[4], Antoine Delhaye[5], Eva Heinz[6], Jean-Francois Collet[5], Adam F. Cunningham[2], Mark A. Blaskovich[1], Robert G. Parton[1,7], Jeff A. Cole[2], Manuel Banzhaf[2], Shu-Sin Chng[3], Waldemar Vollmer[4], Jack A. Bryant[2], Ian R. Henderson[1]*

1 Institute for Molecular Bioscience, The University of Queensland, St. Lucia, Australia, 2 Institute of Microbiology and Infection, University of Birmingham, Birmingham, United Kingdom, 3 Department of Chemistry, National University of Singapore, Singapore, 4 Centre for Bacterial Cell Biology, Biosciences Institute, Newcastle University, Newcastle upon Tyne, United Kingdom, 5 de Duve Institute, Université Catholique de Louvain, Brussels, Belgium, 6 Departments of Vector Biology and Clinical Sciences, Liverpool School of Tropical Medicine, Liverpool, United Kingdom, 7 Centre for Microscopy and Microanalysis, The University of Queensland, St. Lucia, Australia

* e.goodall@uq.edu.au (ECAG); i.henderson@uq.edu.au (IRH)

**Data Availability Statement:** All DNA sequencing data are available from the NCBI BioProjects database (accession number PRJEB43420).

## Abstract

The cell envelope is essential for viability in all domains of life. It retains enzymes and substrates within a confined space while providing a protective barrier to the external environment. Destabilising the envelope of bacterial pathogens is a common strategy employed by antimicrobial treatment. However, even in one of the best studied organisms, *Escherichia coli*, there remain gaps in our understanding of how the synthesis of the successive layers of the cell envelope are coordinated during growth and cell division. Here, we used a whole-genome phenotypic screen to identify mutants with a defective cell envelope. We report that loss of *yhcB*, a conserved gene of unknown function, results in loss of envelope stability, increased cell permeability and dysregulated control of cell size. Using whole genome transposon mutagenesis strategies, we report the comprehensive genetic interaction network of *yhcB*, revealing all genes with a synthetic negative and a synthetic positive relationship. These genes include those previously reported to have a role in cell envelope biogenesis. Surprisingly, we identified genes previously annotated as essential that became non-essential in a Δ*yhcB* background. Subsequent analyses suggest that YhcB functions at the junction of several envelope biosynthetic pathways coordinating the spatiotemporal growth of the cell, highlighting YhcB as an as yet unexplored antimicrobial target.

**Funding:** ECAG was supported by a Biotechnology and Biological Sciences Research Council scholarship to ECAG as part of the Midlands Integrative Biosciences Training Partnership IRH. The research was supported by a European Union's Horizon 2020 research and innovation programme under the Marie Skłodowska-Curie grant agreement No 721484 awarded to WV and IRH. B.Z. was supported by a CSC-UQ PhD Scholarship (China Scholarship Council) to MB. This work was also supported by the National Health and Medical Research Council of Australia (grants APP1140064 and APP1150083 and fellowship APP1156489 to R.G.P.). RGP is supported by the Australian Research Council (ARC) Centre of Excellence in Convergent Bio-Nano Science and Technology CE140100036. This work was supported by the University of Queensland funding to IRH. The funders had no role in study design, data collection and analysis, decision to publish, or preparation of the manuscript.

**Competing interests:** The authors have declared that no competing interests exist.

## Author summary

All life depends on a cell envelope to enclose the chemical reactions that make life possible. But how do cell envelopes grow? How each component of the cell envelope is incorporated into the envelope at the correct amount, in the correct place, and at the correct time, to prevent cell death, has been a long-standing question in bacteriology. Using a unique combination of high throughput chemical genetic screens we identified *yhcB*, a conserved gene of unknown function, required for the maintenance of cell envelope integrity in *Escherichia coli*. Loss of YhcB results in aberrant cell size driven by the production of excess membrane phospholipids. Subsequent molecular and biochemical analyses suggest YhcB influences the spatiotemporal biogenesis of LPS, peptidoglycan and membrane phospholipids. Our data indicate YhcB is a key regulator of cell envelope growth in Gram-negative bacteria playing a crucial role in coordinating cell width, elongation, and division to maintain cell envelope integrity.

## Introduction

The bacterial cell envelope plays a fundamental role in protection, host interaction, energy generation, expulsion of toxic substances and coordination of growth and cell division. As a physical barrier, it has a central role in acquired and intrinsic antimicrobial resistance. Therefore, the systems required for cell envelope biogenesis are important drug targets. In Gram-negative bacteria, the tripartite cell envelope is composed of the cytoplasmic membrane (CM), the peptidoglycan (PG) layer within the periplasmic space and an outer membrane (OM) [1]. The OM is an asymmetrical bilayer of phospholipids and lipopolysaccharide (LPS), which forms a strong permeability barrier conferring resistance to many toxic antimicrobials. Both membranes are studded with integral membrane proteins and peripheral lipoproteins that facilitate cellular functions [2–4]. Each component of the cell envelope must be synthesised and assembled in a coordinated fashion to maintain cell envelope homeostasis and viability. Thus, understanding how this complex envelope is synthesised and maintained is instrumental to understanding how to disrupt its function, to kill the bacterium, or render the organism susceptible to otherwise ineffective treatments.

Over the last 50 years a variety of sophisticated and complex multiprotein machineries have been discovered that are required for the synthesis of the Gram-negative cell envelope. The Lpt machinery facilitates insertion of LPS into the OM [5–10], the BAM complex coordinates OM protein assembly and insertion [11–15], the Lol system traffics lipoproteins across the periplasm for incorporation into the OM [16–20], the Mla complex enables phospholipids transport between the two membranes [21–28], and the elongasome and divisome are the architects of peptidoglycan assembly during cell elongation and division [29–31]. Each pathway in isolation is broadly understood. However, the precise molecular events that govern the crosstalk between the different biosynthetic pathways, both spatially and temporally during growth and cell division, remain to be elucidated.

Genetic screens to identify synthetically lethal interactions and suppressor mutations have played a significant role in identifying the interconnected pathways of envelope biosynthesis [32,33]. Transposon insertion sequencing (TIS) is one approach that enables the study of genetic interactions on a whole genome scale [34]. Here we used a chemical genomics approach to identify genes required for cell envelope homeostasis in *Escherichia coli*, coupling TraDIS, a TIS method [35,36], with the membrane targeting antibiotic polymyxin B. We demonstrate that a poorly characterized protein, YhcB, is crucial for tolerance to polymyxin B and

for maintenance of cell envelope homeostasis in *E. coli*. Previous studies revealed that YhcB is a CM protein widely conserved in Gammaproteobacteria that physically interacts with the elongasome, the machine that coordinates peptidoglycan synthesis along the cylindrical part of the cell, and with LapA, a protein involved in regulating LPS biosynthesis [29,33,37–41]. More recent studies have shown that loss of YhcB results in a more permeable cell envelope, increased sensitivity to cell wall-targeting antibiotics and irregular cell morphologies [42,43]. We demonstrate that loss of YhcB results in dysregulation of cell length and width and that this defect is growth rate dependent. We created a high-density transposon mutant library in an *E. coli* Δ*yhcB* strain to gain a whole genome view of genetic interactions with *yhcB*. Subsequent screening of this library against chemical stresses revealed all mutations within this library that suppress the envelope defects associated with the loss of *yhcB*. These data suggest that YhcB is positioned at the interface between PG and LPS synthesis, and phospholipid membrane biogenesis, and has a role in coordinating the spatial and temporal assembly of the cell envelope.

## Results

### Identification of envelope barrier-defective mutants

Previously, we described a high-density transposon-mutant library of *E. coli* K-12 [44]. To identify genes required for cell envelope biogenesis we applied a chemical genomics approach exposing our library to sub-inhibitory concentrations of the membrane-acting antibiotic polymyxin B (S1A Fig). Polymyxins bind the lipid A moiety of LPS, displacing divalent cations, disrupting the integrity of LPS-crosslinks, and resulting in an increase in membrane permeability [45–48]. We posited that if the transposon disrupts a gene required for the maintenance of cell envelope integrity, the resulting mutant will become more susceptible to polymyxin B, and the gene would be identified through our TIS screen as these mutants will be outcompeted during growth; identifiable as a depletion in transposon-insertions. We grew the transposon library in LB broth with or without 0.2 μg/ml polymyxin B, in duplicate, and harvested cells after ~5 generations of growth (Fig 1A). The sequencing yielded a total of >4.2 M reads, estimated to sample >99% of the possible unique insertion sites (S1B Fig). We used the BioTraDIS pipeline for data processing and analysis [49], and mapped the processed reads to the *E. coli* BW25113 reference genome obtained from NCBI (accession CP009273.1). The insertion index scores (IISs; the number of insertions per gene, normalised for gene length) between replicates were highly comparable (S1C Fig). We identified 54 genes required for growth in sub-inhibitory concentrations of polymyxin B (Fig 1B and S1 Table). Comparison of the relative abundance of COG (Cluster of Orthologous Groups) categories of these 54 genes showed a marked increase in genes involved in cell wall/membrane/envelope biogenesis (COG category 'M'), supporting the validity of the TIS screen (S1D Fig; [50] http://eggnog-mapper.embl.de/ date accessed: 19th March 2021). Many of these genes (e.g. *bamB*, *degP*, *galE*, *surA*, and *waaBCFGJPR*) have been identified in a similar TIS screen using *Klebsiella pneumoniae* or have previously been reported as mutants sensitive to polymyxin [51,52]. The COG analysis identified only three genes (*ydbH*, *yqjA* and *yhcB*) of unknown function (category 'S' in S1D Fig) that were important for survival when exposed to sub-inhibitory concentrations of polymyxin B (Figs 1C and S1E). YdbH is a putative autotransporter, while YqjA belongs to the DedA family of proteins that have been suggested to be involved with membrane homeostasis [53]. Given that *yhcB* is conserved in three out of the six ESKAPE pathogens, is required for tolerance to polymyxin, an antibiotic of last resort, and is the target of a sRNA toxin that causes cell death, we decided to investigate the role of YhcB further [40,54].

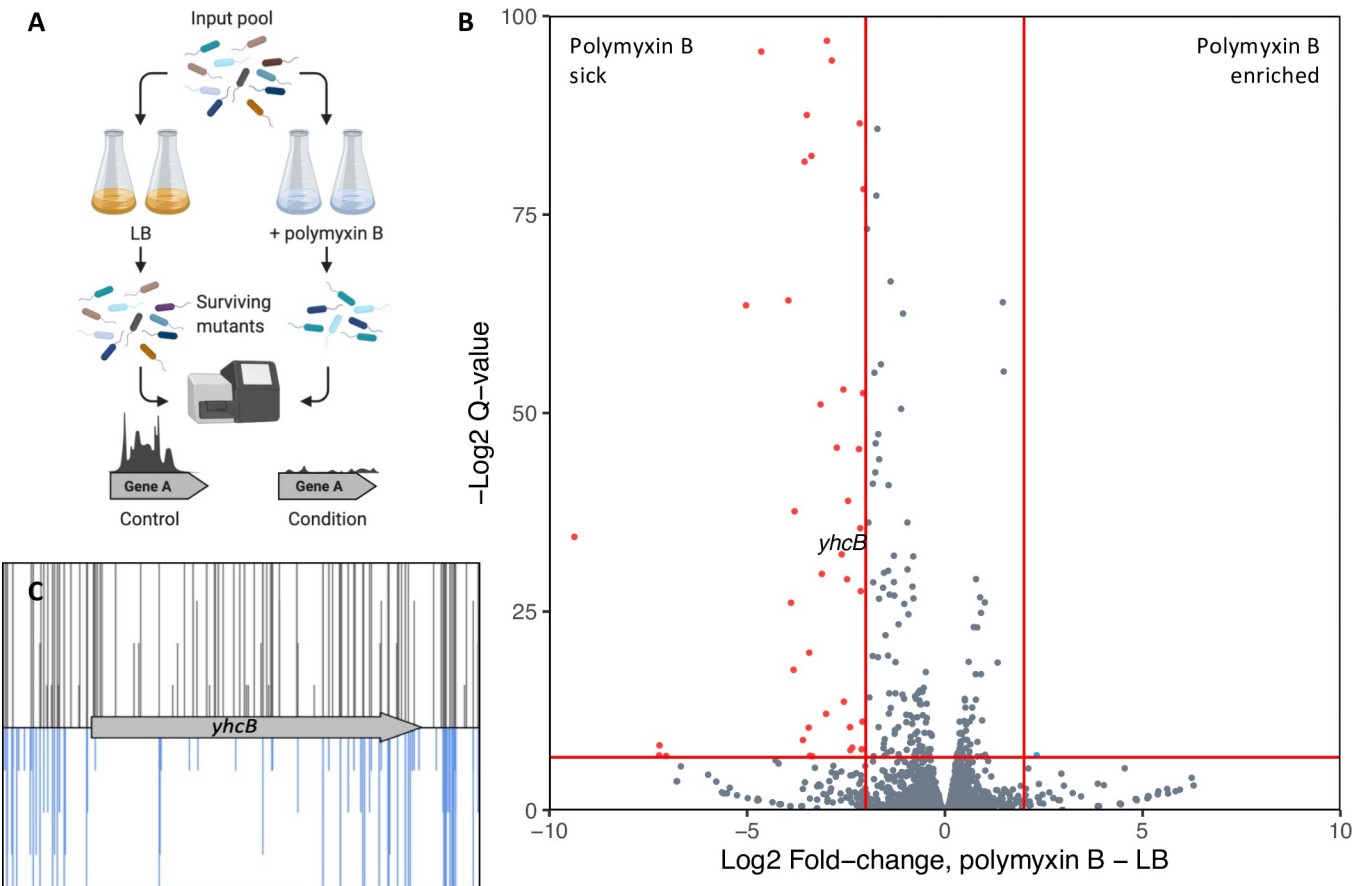

**Fig 1. Screening the *E. coli* BW25113 transposon library in sub-inhibitory concentrations of polymyxin B.** (A) Schematic showing the TIS experiment for identification of polymyxin B-sensitive mutants, created with BioRender.com (B) Volcano plot of the fold-change in mapped reads between conditions, calculated using BioTraDIS. Datapoints in red correspond with genes with >2-fold decrease in reads in the polymyxin B dataset relative to the LB outgrowth dataset, and with a Q-value >0.01 (red horizontal line). A label for the *yhcB* datapoint has been added above the point for clarity. (C) Image of the *yhcB* insertion data following outgrowth in LB only (grey, above) or in LB supplemented with polymyxin B (blue, below). The transposon insertion position along the gene is marked by a vertical line, with the vertical line size corresponding with read depth.

## Growth rate dependent effect on cell size and antibiotic sensitivity

We observed that the two Δ*yhcB* mutants from the Keio library do not phenocopy each other. Whole genome sequencing revealed mutations at secondary sites in both isolates when compared to the parent strain, *E. coli* BW25113. Therefore, we constructed a Δ*yhcB* mutant and confirmed that it was isogenic with the parent strain by whole genome sequencing. To confirm the barrier defect of the *yhcB* mutant, we grew the parent strain, *E. coli* BW25113, and our isogenic Δ*yhcB* mutant on LB agar plates supplemented with either vancomycin, a high molecular weight antibiotic that is typically unable to cross the Gram-negative OM, or plates supplemented with SDS and EDTA [55,56]. The Δ*yhcB* mutant was more sensitive than the parent strain to both conditions (Fig 2Ai). Ectopic expression of YhcB under arabinose induction on a pBAD plasmid was able to complement sensitivity defects (S2A Fig), consistent with previous reports [43]. These data confirm that loss of *yhcB* results in a severe envelope defect.

Despite normalising cultures by optical density, the number of viable colony forming units (CFUs) of the Δ*yhcB* mutant were 10-fold fewer than the parent on the LB control plate (Fig 2Ai). Analysis of cellular morphology by DIC microscopy revealed that the Δ*yhcB* mutant cells were significantly larger than the parent strain when grown in LB (Fig 2B). Expression of *yhcB*

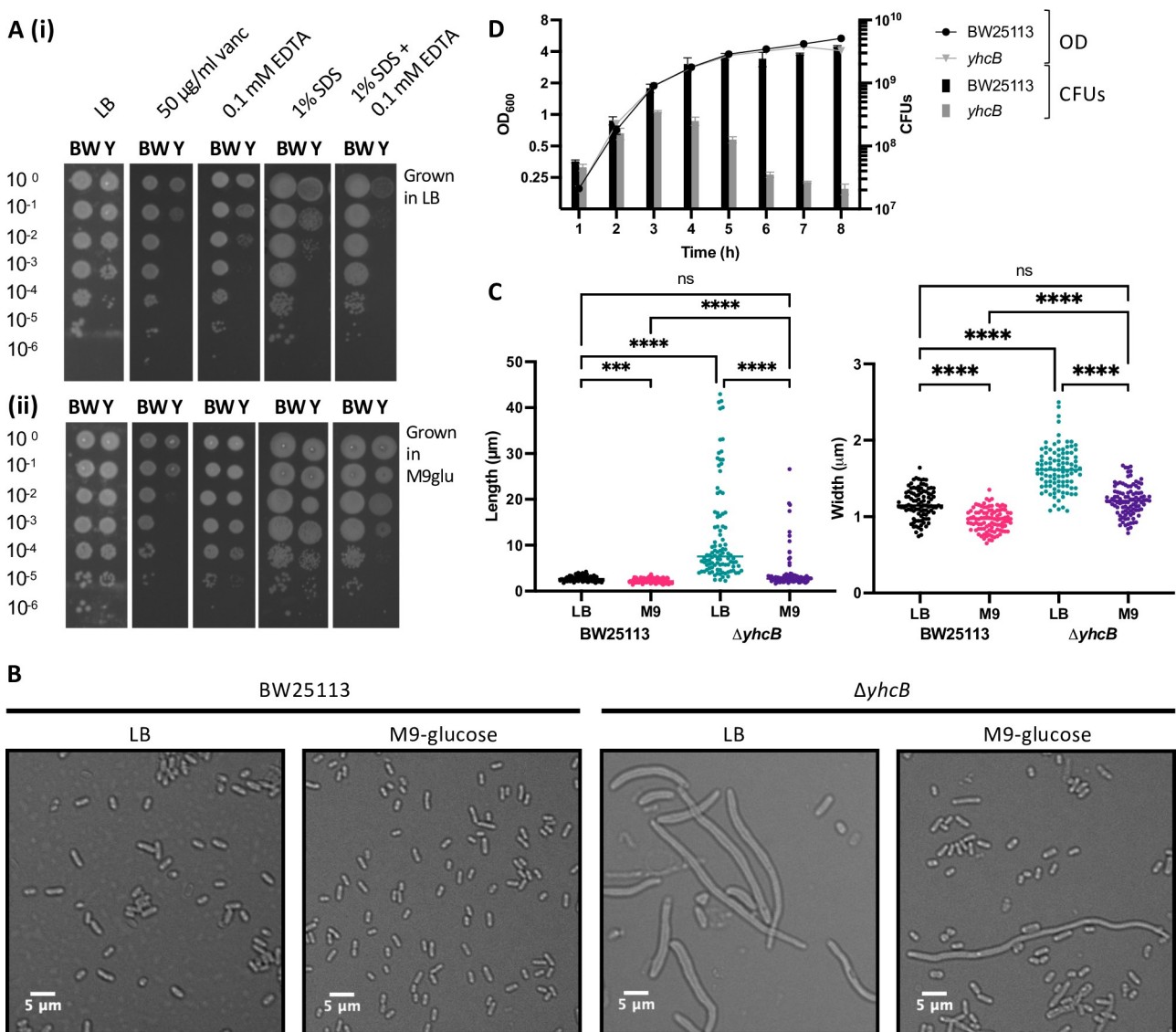

**Fig 2. Validation of a *yhcB* mutant cell envelope defect and the effect of growth medium on cell size.** (A) Overnight cultures of the wild-type strain BW25113 (BW) and Δ*yhcB* (Y) strain grown in LB or M9-glucose at 37˚C, normalised to an $OD_{600}$ of 1.00 and 10-fold serially diluted before inoculating LB agar plates supplemented with various compounds. (B) The wild-type and Δ*yhcB* mutant were grown overnight in LB or M9-glucose media at 37˚C. Cells were visualised using a Nikon 90i microscope, a 5 μm bar is shown for scale. (C) Cell measurements for BW25113 and Δ*yhcB* mutant grown in LB and M9-glucose. Images were taken after overnight growth and are representative of n = 4 experiments, data from 1 experiment is presented. Width and Length measurements were taken of 100 cells in each condition. A Kruskal-Wallis test with Dunn's correction for multiple comparisons was used to compare between samples: $p < 0.0001$ (****); $p < 0.0002$ (***). (D) The optical density (OD) and number of colony forming units (CFUs) were recorded hourly over 8 h during growth in LB at 37˚C. Strains were grown in triplicate, the mean is plotted and the s.d. represented by error bars. BW25113 is shown in black (•), *yhcB* is shown in grey (▼).

in the Δ*yhcB* mutant background substantially decreased the cell dimensions relative to the Δ*yhcB*-pBAD empty vector strain (S2B Fig). We also observed that the ectopic expression of *yhcB* in the *E. coli* BW25113 parent strain resulted in longer cells than the control. Taken together, these data suggest that the abundance of YhcB within the cell impacts cellular dimensions, and that YhcB is important for maintaining cell size and shape.

Bacterial cell dimensions are influenced by a number of factors (reviewed by Cesar and Huang [57]), but cell size is fundamentally determined by growth rate and nutrient availability

[58–61]. A shift from growth in nutrient-restricted to nutrient-rich medium results in an increase in both cell length and cell width [62]. Therefore, we investigated the effect of growth medium on the size of the Δ*yhcB* mutant. Overnight growth in M9 minimal medium supplemented with glucose (M9-glucose) substantially decreased the cell size of the Δ*yhcB* mutant compared to growth in LB, although a small subset of cells within the population still exhibited a division defect and increased cell length (Fig 2B and 2C). Moreover, overnight growth in M9-glucose restored resistance to membrane-acting compounds SDS and EDTA, but cells remained susceptible to vancomycin (Fig 2Aii). Changes in temperature alter growth rate but not size [61]. To understand whether the fitness advantage conferred by growth in M9-glucose was due to a slower doubling time, or nutrient-dependent, we grew the Δ*yhcB* mutant in a range of conditions. Overnight growth in M9-glucose and M9-glycerol at 37˚C, and LB at 16˚C, restored resistance to both SDS and EDTA, but no growth condition conferred resistance to vancomycin (S3 Fig). Altogether these results indicate that a slower growth rate partially alleviates the defect caused by deletion of *yhcB* but does not restore the envelope permeability barrier.

We hypothesised that there is an envelope synthesis defect in a Δ*yhcB* mutant that becomes more severe over time during growth in LB. To investigate this, overnight cultures (grown first in M9 medium) were sub-cultured in LB at a starting optical density (OD) of 0.05. Bacteria were grown at 37˚C with aeration, and both the $OD_{600}$ and the corresponding number of CFUs were recorded hourly. The rate of increase in biomass of the mutant, as assessed by optical density, was comparable to that of *E. coli* BW25113 (Fig 2D). In contrast, upon transition to stationary phase, the number of Δ*yhcB* CFUs decreased 10-fold from ~3.5 x $10^8$ to ~3.5 x $10^7$. These data suggest that the fast-growing mutant cells gradually expand in cell size, fail to divide reliably, and lose viability when they enter the stationary growth phase.

## The domains of YhcB contribute to survival under different conditions

To gain a better understanding of how YhcB functions, we scrutinized the primary structure of the protein. A prediction of the secondary structure of YhcB by PSIPRED supports the model put forward by Mogi *et al.* [40,63,64]. YhcB has a single transmembrane (TM) domain, with a cytoplasmic helical domain and a disordered cytoplasmic C-terminal region (S4A Fig); the topology of which has been confirmed by GFP- and PhoA-fusion analysis [39]. A recent crystal structure was resolved for the TM and cytoplasmic helical domains and corroborates the multiple lines of evidence that YhcB self-interacts [37,39,42]. A multisequence alignment of the amino acid sequence of YhcB from 150 different species was generated using the online program ConSurf [65]. This revealed two highly conserved amino acid motifs within the disordered C-terminus: a 'NPF' motif after the long cytoplasmic helical domain, followed by a non-conserved linker region, and a highly conserved 'PRDY' motif at the extreme C-terminus (S4B Fig). A large number of transposon insertions were identified within the 3′ end of *yhcB* in the initial polymyxin TIS screen; the *yhcB* gene only met the stringency criteria for identification when 20% of the 3′ end of the coding sequence (CDS) was discarded. We mapped the coordinates of these domains to the original TIS data and the data suggest that the 'PRDY' domain is dispensable for survival in the presence of polymyxin B (Fig 3A). To confirm the functional importance of the 'PRDY' domain, which is highly conserved (S5 Fig), we constructed complementation vectors encoding truncated variants of YhcB and repeated our plate-based envelope screens (Fig 3B and 3C). Consistent with results reported by Sung *et al.* [43], the ΔTM construct fully complemented both phenotypes. However, deletion of the 'PRDY' domain rendered the mutant sensitive to vancomycin, but not SDS-EDTA (Fig 3C). These data suggest a dual function of the YhcB protein: the cytoplasmic helical domain is needed to suppress

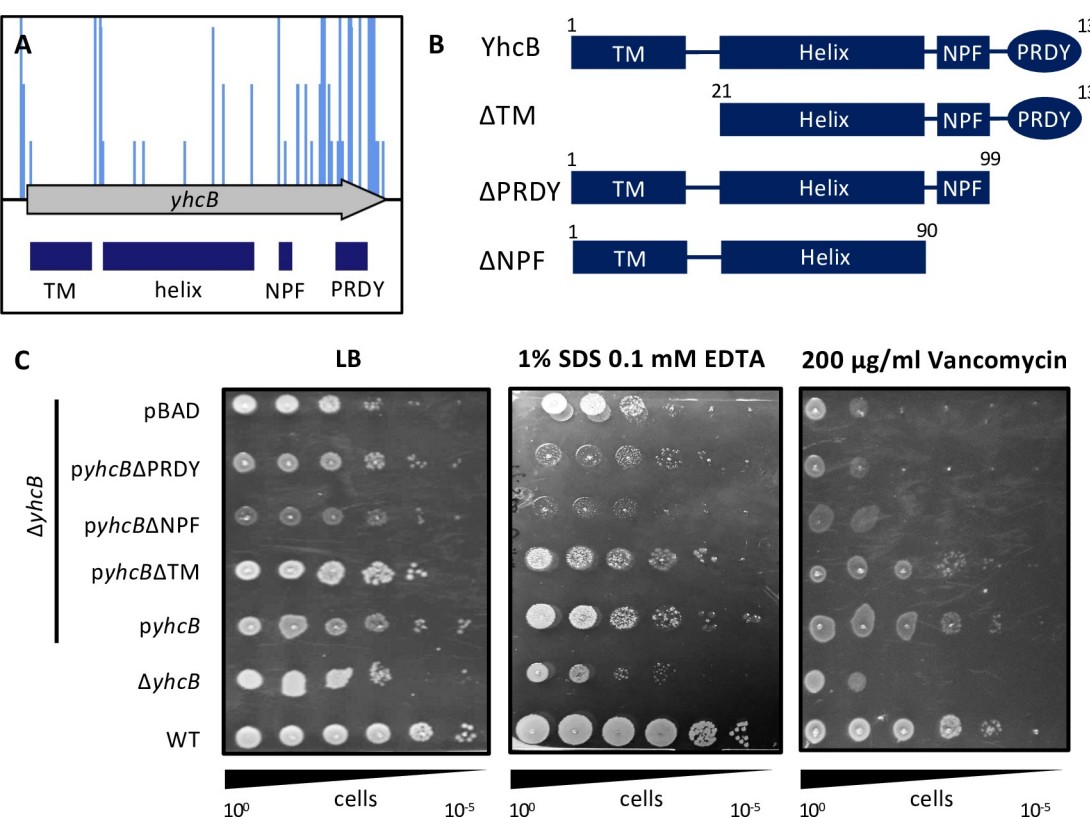

**Fig 3. The structural and functional contribution of the domains of YhcB.** (A) Polymyxin B TIS screen data (blue insertions) plotted above the gene track with the domains of YhcB mapped underneath (dark blue boxes). Insertion sites are represented by vertical bars and are capped at a frequency of 3. (B) Schematics of the plasmid-based YhcB complementation constructs. (C) Overnight cultures grown in LB supplemented with carbenicillin (for plasmid selection), normalised to an $OD_{600}$ of 1.00 and 10-fold serially diluted before inoculating LB agar plates supplemented with vancomycin or SDS and EDTA, with 0.2% arabinose. Abbreviations: TM = transmembrane domain. 'NPF' and 'PRDY' are conserved amino acid sequences.

sensitivity to SDS and EDTA, while the function of the 'PRDY' domain is critical for survival in the presence of vancomycin, but not polymyxin B or SDS and EDTA. Further work is needed to characterise the role of the 'PRDY' domain, and the structure of this region remains elusive. However these data suggest the function of the 'PRDY' domain is condition-specific as it is critical for survival in the presence of vancomycin, but not polymyxin B or SDS and EDTA. Given the highly conserved nature of this domain across multiple species, it is our hypothesis that this domain is functionally important and is functionally distinct from the cytoplasmic helical domain.

## Identification of genetic interactions with *yhcB*

To identify genes with a synthetically lethal relationship with *yhcB*, a transposon mutant library was constructed in a Δ*yhcB* background using a mini Tn*5* transposon encoding a kanamycin resistance gene. A second library was constructed in the *E. coli* BW25113 parent strain using the same transposon as a control. Approximately 800,000 transposon-mutants were collected and pooled for each library. Two technical replicates of each library were sequenced (S6A Fig and S2 Table). There was a high correlation coefficient between the gene insertion index scores of each replicate for both libraries ($R^2 = 0.93$ and $R^2 = 0.96$ for the wild-type and Δ*yhcB* library respectively). The insertion sites were evenly distributed around the genome

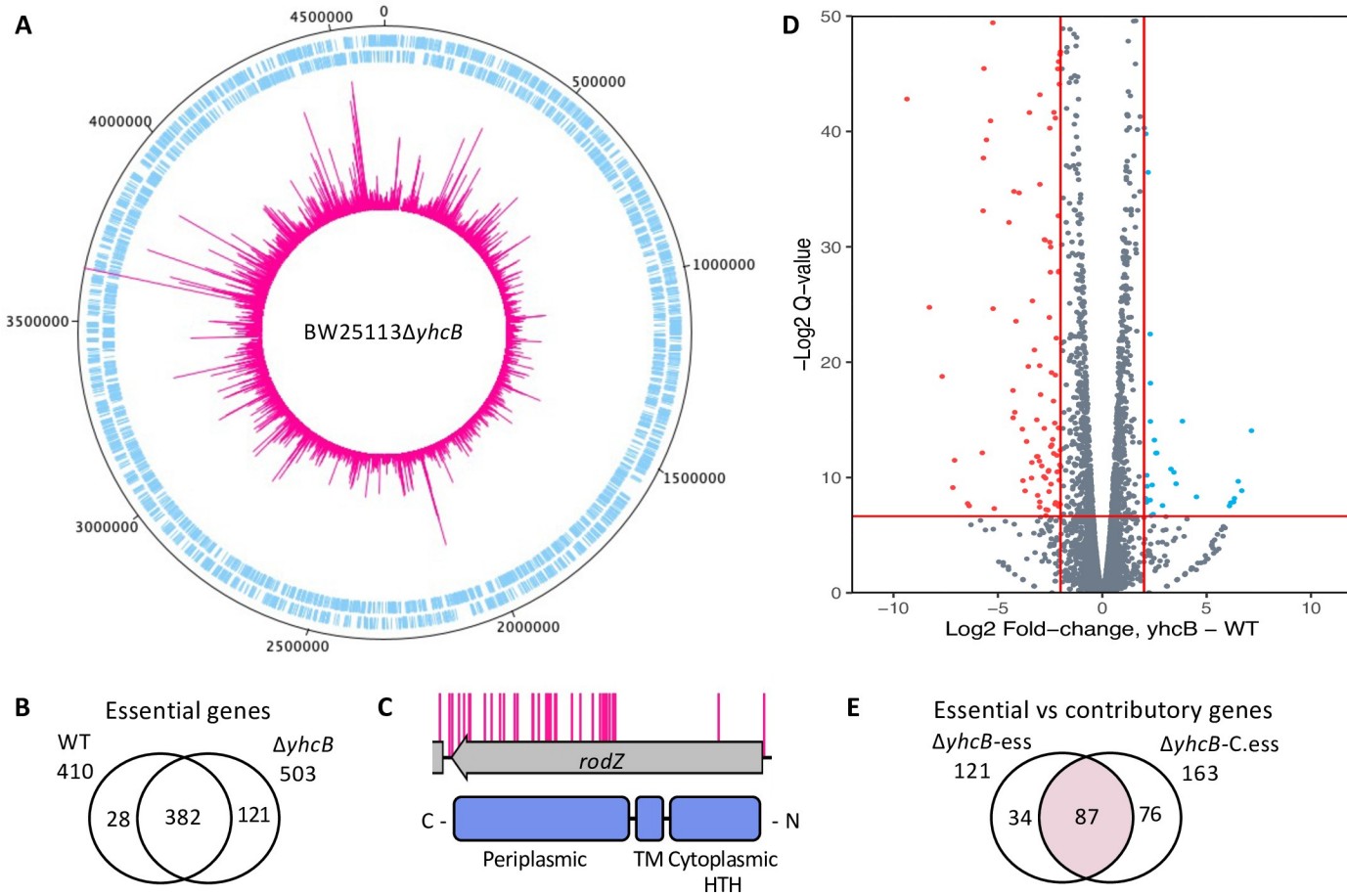

**Fig 4. Construction of a transposon library in a *yhcB* mutant and identification of genes with a synthetic lethal relationship with *yhcB*.** (A) The Δ*yhcB* transposon library. A genome map of BW25113 starting at the annotation origin with the sense and antisense coding sequences of BW25113 shown in blue, respectively, and the position and frequency of sequenced transposon insertion events shown in pink. (B) Comparison of genes identified by bimodal analysis as significantly underrepresented by transposon mutants in each library. (C) The insertion profile of *rodZ* in the Δ*yhcB* library, with RodZ domains shown below in blue. (D) The log$_2$ fold-change of read depth per gene between the parent BW25113 and the Δ*yhcB* transposon library. Genes enriched for transposon insertions (>2-fold) in the *yhcB* library are shown in blue, indicative of more fit mutants. Genes with a >2-fold decrease in insertions are shown in red, indicative of sick mutants. A Q-value threshold of 0.01 was used (red horizontal line) (E) Comparison of genes predicted to be essential (sparsely disrupted genes) with genes identified as having a >2-fold decrease in reads compared to the control library (genes that contribute to fitness). This is a comparison of the bi-modal and enrichment analyses outputs. The overlap of these analyses are the 87 identified synthetic lethal genes (pink). Abbreviations: wild-type (WT); transmembrane domain (TM); helix-turn-helix domain (HTH).

(Figs 4A and S6B), and the respective insertion density amounted to an average of one insertion every 6.28 bp in the wild-type library, and one insertion every 6.99 bp in the Δ*yhcB* library. We quantified insertions per CDS to identify genes that were sparsely disrupted by transposon insertion events i.e. essential genes in each genetic background [49]. There was an overlap of 382 essential genes between the two datasets and these were not considered further (Fig 4B).

Previously, Li *et al.* [37] validated that the deletion of both *rodZ* and *yhcB* is lethal [66]. RodZ is a bitopic CM protein that couples peptidoglycan synthesis to the MreB cytoskeleton [67]. The synthetic lethal relationship of *rodZ* and *yhcB* genetically links the elongasome with the function of YhcB. Our data confirmed a synthetic lethal interaction between *rodZ* and *yhcB*. However, we observed that only the 5′ end of the *rodZ* gene, corresponding with the cytoplasmic helix-turn-helix and transmembrane domains of the RodZ protein, is essential in a *yhcB* background (Fig 4C). These domains of RodZ are required for maintaining cell shape

[67,68]. This result supports the validity of our approach. Surprisingly, our bimodal analysis of the data revealed 28 genes that were predicted to be essential in a wild-type library but no longer essential when *yhcB* is deleted (Fig 4B and S3 Table). This list includes *ftsH* (S6C Fig), which encodes a metalloprotease that degrades several proteins, including LpxC. Deletion of FtsH results in toxic accumulation of LPS via increased LpxC stability, which is lethal to the cell [32]. The increased number of transposons in *ftsH* in the Δ*yhcB* library suggest that an increase in LpxC stability in this background is less toxic. To support this observation, we deleted *ftsH* in our Δ*yhcB* strain via Datsenko-Wanner recombination (S6D Fig) [69].

The bimodal analysis revealed 121 genes that were predicted to be essential in a Δ*yhcB* background. However, this analysis pipeline does not identify the non-essential genes that contribute to fitness: such mutants are viable but either more or less fit. Consequently, when compared to the control library they are represented by higher or lower numbers of sequencing reads, respectively. Therefore we further analysed our data using the tradis_comparison.R script [49]. This analysis uses edgeR and measures the fold-change of sequence read depth between a condition and control [70]. The 382 genes reported to be essential in both libraries were removed from this analysis. After filtering our data, this method identified 22 genes that, when disrupted, confer a fitness advantage (Fig 4D, blue), and 163 genes that, when disrupted, confer a fitness defect in a Δ*yhcB* mutant (Fig 4D, red and S4 Table; Fold change > 2, Q-value < 0.01). By combining these analyses, we defined synthetically lethal genes as those that are both significantly underrepresented by transposon insertion mutants and that have a >2-fold decrease in reads when compared to the control with a Q-value < 0.01. Altogether 87 genes met these criteria: we consider these genes to be synthetically lethal with *yhcB* and they are discussed below (Fig 4E).

## Identification of pathways important for viability of a *yhcB* mutant

As YhcB interacts directly with components of the elongasome, and is synthetically lethal with *rodZ*, we inspected the list of synthetic lethal genes for additional genes involved in peptidoglycan synthesis, remodelling or recycling. Five genes (*mepM*, *mepS*, *dacA*, *dapF* and *ldcA*) were identified (Fig 5A). The genes *mepM* and *mepS* encode two of the three peptidoglycan endopeptidases that cleave the 3–4 *meso*-Dap-D-Ala crosslink and are collectively essential for cell elongation in *E. coli* [71]. However, a third endopeptidase of this group encoded by *mepH*, was not essential in a Δ*yhcB* mutant (Fig 5A and 5B). DapF is an epimerase that catalyses the conversion of L, L-diaminopimelate (LL-DAP) to *meso*-diaminopimelate (*meso*-DAP), which is an integral component of the peptidoglycan stem peptide and the primary residue from which cross-links are formed [72]. Sacculi isolated from *dapF* mutants have fewer crosslinks [73]. The DD-carboxypeptidase PBP5, encoded by *dacA*, removes the terminal D-alanine from peptidoglycan pentapeptides [74], and is the primary carboxypeptidase under standard laboratory growth conditions [75,76]. Finally, the cytosolic LD-carboxypeptidase LdcA participates in the recycling of peptidoglycan turnover products (Fig 5A). Deletion of *ldcA* results in the lysis of cells during entry into the stationary growth phase due to the accumulation of peptidoglycan crosslinking defects [77]. From these data, it could be inferred that a Δ*yhcB* mutant cannot survive when additional mutations weaken the integrity of the peptidoglycan layer.

To determine whether other cellular processes or pathways are enriched among the genes identified as synthetically lethal with *yhcB*, we used the PANTHER overrepresentation test [78]. This analysis compares the identified proportion of genes in a given functional category to the expected number of genes in a pathway derived from whole genome data (S5 Table). Four processes were identified as functionally enriched among the list of synthetic lethal genes: the 'enterobacterial common antigen (ECA) biosynthetic pathway', 'LPS core

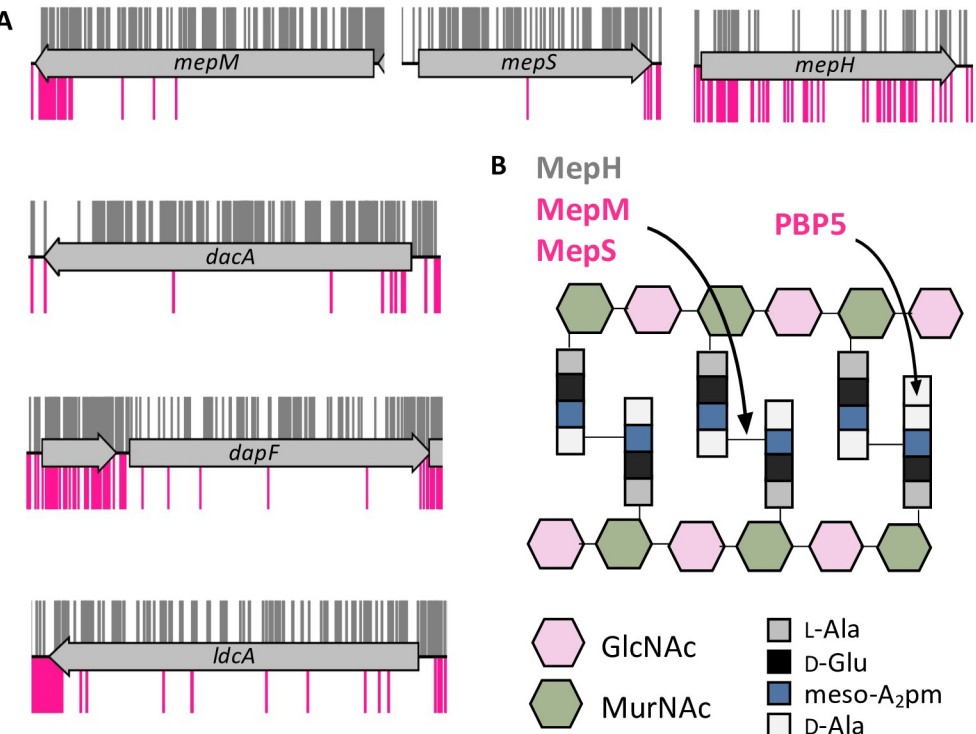

**Fig 5. _yhcB_ is synthetically lethal with components of PG synthesis and recycling pathways.** (A) TIS data of the BW25113 library (grey) and BW25113Δ_yhcB_ library (pink) were plotted above and below the gene track, respectively. Insertion sites are represented by vertical bars and are capped at a frequency of 1. (B) Schematic representation of the target sites of the endopeptidases and PBP5 (encoded by _dacA_). Abbreviations: _meso_-diaminopimelate (_meso_-A$_2$pm); N-acetylglucosamine (GlcNAc); N-acetylmuramic acid (MurNAc); L-alanine (L-ala); D-glutamic acid (D-Glu); D-alanine (D-Ala).

biosynthesis', 'cell division', and 'response to abiotic stimulus' (Fig 6A). With the exception of the _tol-pal_ system (S7A Fig), the remaining genes listed within the 'cell division' and 'response to abiotic stimulus' categories did not share overlapping functions and we did not pursue these further.

Upon closer inspection of LPS biosynthetic pathways, both the heptosyl transferases WaaC and WaaF, and the entire heptose biosynthetic pathway were synthetically lethal (Figs 6B,6C, and S7B). The WaaP kinase, which phosphorylates the first heptose of the inner core, was also essential in a _yhcB_ mutant. Enzymes WaaBGQRY were not essential in a Δ_yhcB_ background, but mutants were less fit. In contrast, ligation of the third 2-Keto-3-deoxy-octonate (Kdo) moiety by WaaZ was non-essential. The acylation of lipid A by LpxM was also essential in a Δ_yhcB_ strain (Fig 6C). Together these data suggest that OM rigidity or structural integrity, whether mediated by Lipid-A hexa-acylation or LPS crosslinking interactions, is important for viability of a Δ_yhcB_ strain.

Within the locus required for ECA biosynthesis the genes _wecCDEF_, _rffH_ and _wzxE_ were synthetically lethal with _yhcB_, while disruption to _wecB_ and _wecG_ suggests these genes were not synthetically lethal but the mutants were sick (Figs 6D and S7C). There were no significant differences in the insertion frequency within genes _wzzE_ and _wecA_. WzzE mediates the length of the ECA chains by determining the number of repeating units [79], and WecA catalyses the first committed step in ECA biosynthesis: the transfer of _N_-acetylglucosamine-1-phosphate onto undecaprenyl phosphate (Und-P) to form ECA-lipid I. In short, an ECA-null mutant is

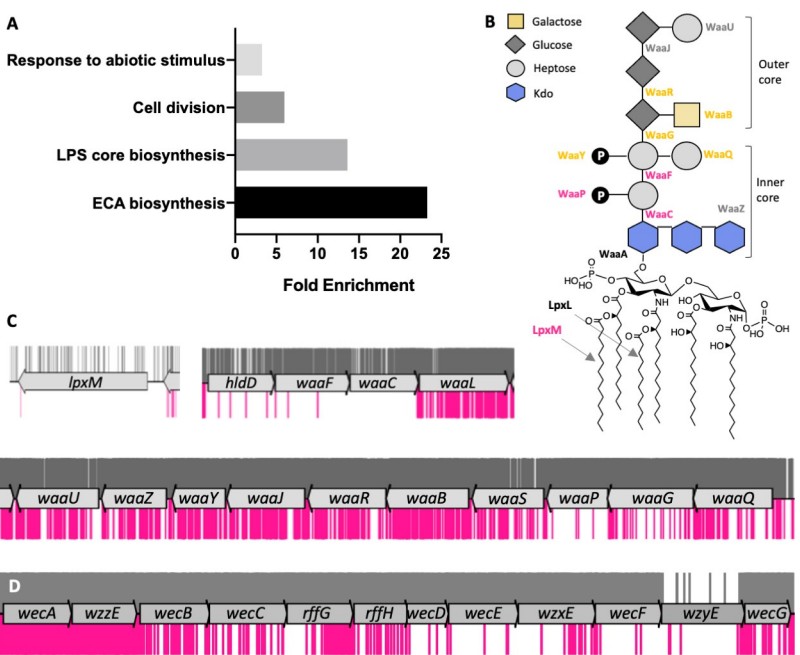

**Fig 6. Functional enrichment among synthetic lethal genes.** (A) Functional enrichment of cellular processes among the genes that are synthetically lethal with *yhcB*, with Fisher's exact test with Bonferroni correction for multiple testing. Results displayed for P < 0.05. (B) A schematic of the structure of LPS. LPS biosynthesis enzymes are indicated next to the linkage they form (central column) or component they ligate (side branches). Synthetic lethal enzymes are labelled in dark pink, enzymes that contribute to fitness are labelled in orange, core essential enzymes in black and non-essential enzymes in grey. (C) TIS data for BW25113 (grey) and BW25113Δ*yhcB* (pink) libraries were plotted above and below the gene track, respectively. Insertion sites are represented by vertical bars and are capped at a frequency of 1. Insertions within genes required for LPS core biosynthesis, and *lpxM*, are significantly underrepresented. (D) The insertion profiles in genes of the enterobacterial common antigen biosynthetic pathway. Abbreviations: Lipopolysaccharide (LPS); Enterobacterial Common Antigen (ECA).

viable and mutants with variable ECA lengths are also viable; only mid-pathway blocks are lethal. Mutations that introduce a mid-pathway block in ECA biogenesis are known to cause an aberrant cell morphology [80], due to accumulation of undecaprenyl-linked ECA-lipid II intermediates creating limited availability of Und-P: a compound of limited abundance that sits at the start of several cell envelope biosynthetic pathways including peptidoglycan [80]. These observations suggest that the combined defects of *yhcB*-deletion and Und-P sequestration are lethal, and indicate that the availability of Und-P might be limited in a *yhcB* mutant.

In support of our TIS data, we attempted to construct double deletion mutants by P1 transduction. We selected a representative gene (or genes) required for the synthesis of the following components of the envelope: peptidoglycan, LPS and ECA. We were unable to transduce the *dacA*, *mepS*, *lpxM* or *wecF* mutations into the Δ*yhcB* strain, consistent with our TIS data (S7D Fig). As these cultures were grown in LB and normalised by OD, consistent with our earlier observations, the Δ*yhcB* mutant has an approximate ~10-fold decrease in viable cells when compared to the positive control.

## Identification of mutations that suppress a *yhcB*-deletion defect

In addition to synthetic lethal interactions, mutations that restore a phenotypic defect, so-called suppressor mutations, can assist in the identification of interconnected cellular pathways [32]. We applied TIS to identify, at a whole genome scale, mutations that can restore tolerance

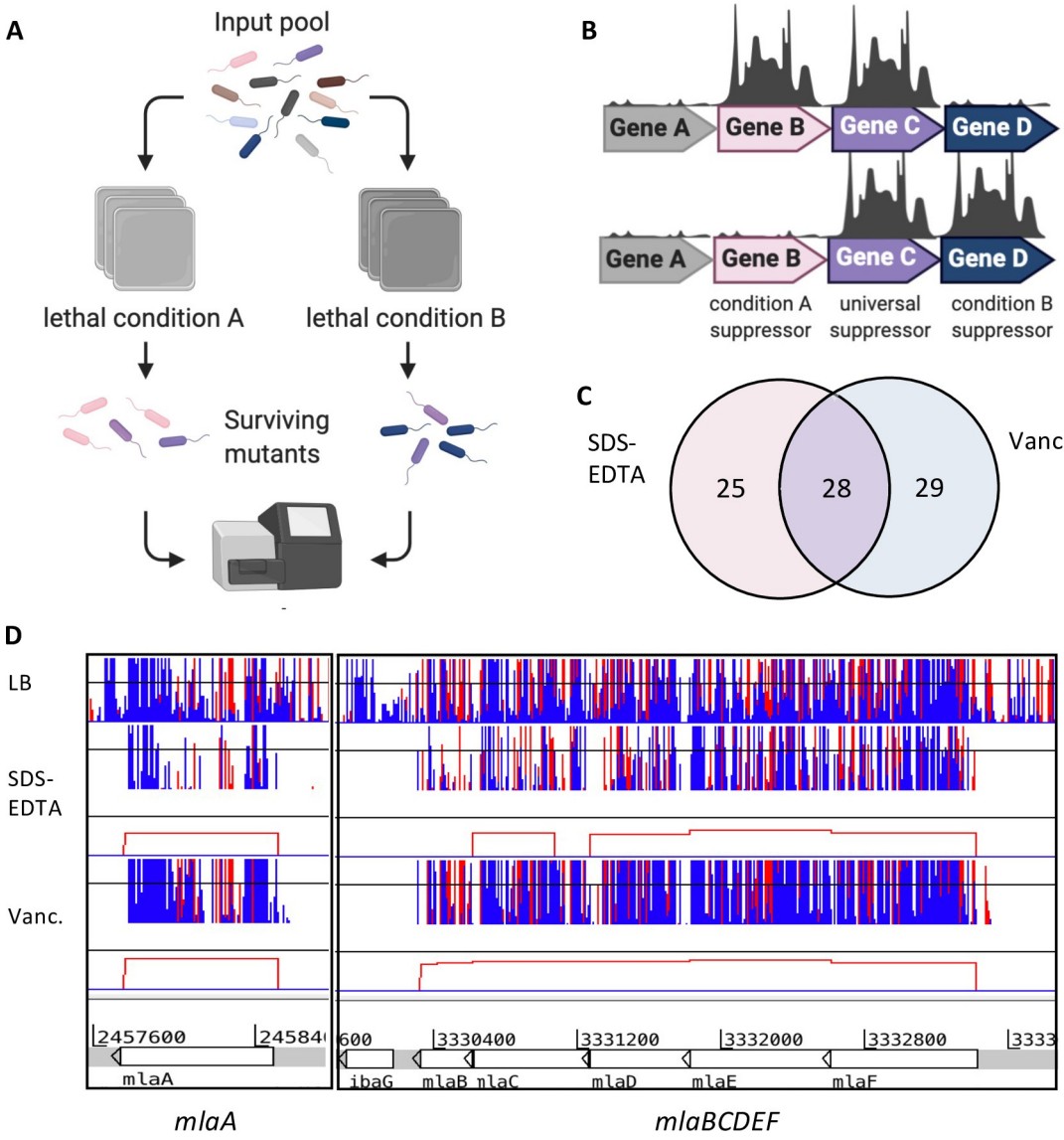

**Fig 7. Identification of suppressor mutations that restore a *yhcB*-deletion defect.** (A) Experimental overview. (B) Cartoon of output. (C) Overlap between gene-deletion suppressors identified in the vancomycin and SDS + EDTA screens. (D) Transposon insertion data for suppressor screens. Red and blue lines indicate the transposon insertion position corresponding with the transposon orientation at the point of insertion. The height of the bar corresponds with mapped sequencing read frequency. The red boxes underneath each suppressor dataset represent significant differential abundance of insertions of the condition compared to the control, identified by AlbaTraDIS. Panel A and B schematics created with BioRender.com, Panel D insertion plots visualized in Artemis [130].

to vancomycin or SDS and EDTA, and therefore suppress a Δ*yhcB* phenotype. Targeted validation experiments would be needed to distinguish specific from non-specific suppressor mutations, but the TIS data can highlight functions that are restorative in a *yhcB* mutant for further investigation. First, the Δ*yhcB* mutant library was screened on LB agar supplemented with either SDS and EDTA or vancomycin at concentrations that kill the Δ*yhcB* strain. Transposon mutants of the Δ*yhcB* library able to grow under these conditions were identified as before (Fig 7A and 7B). We used the recently published AlbaTraDIS package to analyse our data [81]. We identified 28 "knockout" suppressor mutations shared between both conditions, and therefore

considered these to be universal suppressors (Fig 7C and S6 Table). Of note among the 28 universal suppressors was the Mla pathway. All genes of the pathway have insertions along the full length of each CDS indicating a disruption at any stage of the pathway is restorative to both vancomycin and SDS and EDTA sensitivity (Fig 7D). This was unexpected as *mla* mutants are highly sensitive to SDS and EDTA [23,82,83].

Deletion of *nlpI* was also identified as a universal suppressor (S8A Fig). NlpI is an OM lipoprotein that functions as an adaptor protein for peptidoglycan endopeptidases and mediates the degradation of MepS by the protease Prc [84–86]. In a Δ*nlpI* strain, MepS activity is significantly increased [85], which has been shown to enhance peptidoglycan synthesis by stimulating PBP1B-mediated peptidoglycan synthesis and directing peptidoglycan precursors away from the elongasome complex [87], presumably to facilitate the repair of defects in the PG [88]. Consistent with this hypothesis, *mrcB* (PBP1B) was identified in the list of genes that confer a fitness defect on a *yhcB* mutant strain when deleted (S4 Table), while deletion of *mrcA* (PBP1A) and *lpoA*, which encodes the PBP1A activator, were identified as universal suppressors. Suppressor analysis therefore supported a functional link between YhcB and PG synthesis.

Another universal suppressor we identified was deletion of *fabF*, genetically linking *yhcB* with fatty acid biosynthesis (S8B Fig). FabF, together with FabB, are the two β-ketoacyl-[acyl carrier protein] synthases involved in fatty acid elongation [89]. FabB is the major synthase and is essential while FabF is predominantly required at low temperatures to increase membrane fluidity by increasing the proportion of diunsaturated phospholipids [90], as FabF is more efficient than FabB at elongating palmitoleic acid ($16:1\Delta9$) to cis-vaccinic acid ($18:1\Delta11$). Therefore, loss of *fabF* might result in decreased membrane fluidity or a decreased rate of phospholipid synthesis, or both.

We constructed double deletion mutants to validate these findings. Deletion of *mlaD*, *nlpI* or *fabF* restored resistance of a Δ*yhcB* mutant to both SDS + EDTA and vancomycin, confirming these mutations as dominant suppressors able to restore the defect of a Δ*yhcB* mutant under two different growth conditions (S8C Fig). Of note, the double Δ*yhcB*Δ*mlaD*::*aph* mutant was more resistant to SDS + EDTA than either single mutation (S8C Fig).

We plotted the total suppressor insertion data on a genome map to view relative mutant abundance; the assumption here is that read depth is representative of mutant abundance, which correlates with fitness and is an indicator of the strength of the suppression. In addition to Δ*fabF*, disruption of the *mla* operon and Δ*nlpI* we observed a substantial peak at the *uppS-cdsA* locus in both conditions (Fig 8A). UppS, encoded by *uppS* (*ispU*), is an undecaprenyl pyrophosphate synthase, which synthesises undecaprenyl diphosphate (Und-PP), the only source of *de novo* synthesised Und-PP and the precursor to Und-P [91,92]. CdsA sits before the branchpoint in the synthesis of the major phospholipids and catalyses the synthesis of cytidine diphosphate-diacylglycerol from phosphatidic acid [93,94]. Both UppS and CdsA are essential, as such, neither gene can be disrupted by transposon insertion. However, we observed that disruption upstream of the *uppS-cdsA* operon is significant in both suppressor conditions (Fig 8B). It was unclear from the insertion pattern the effect that disruption at this locus would have. However, these data suggest that deregulation of the native level of expression of *uppS-cdsA* is restorative in a *yhcB* mutant background.

During screening of the transposon mutant library on supplemented agar plates, we also isolated six colonies of the parent Δ*yhcB* strain growing on the control plates of LB supplemented with vancomycin, assumed to be natural revertant suppressors of Δ*yhcB*. The revertant suppression was confirmed by plating on both vancomycin and SDS-EDTA, as well as confirming gene-deletion by PCR of the *yhcB* locus (S9A and S9B Fig). We sequenced the genomes of these isolates and identified the mutations listed in S7 Table. Two isolates

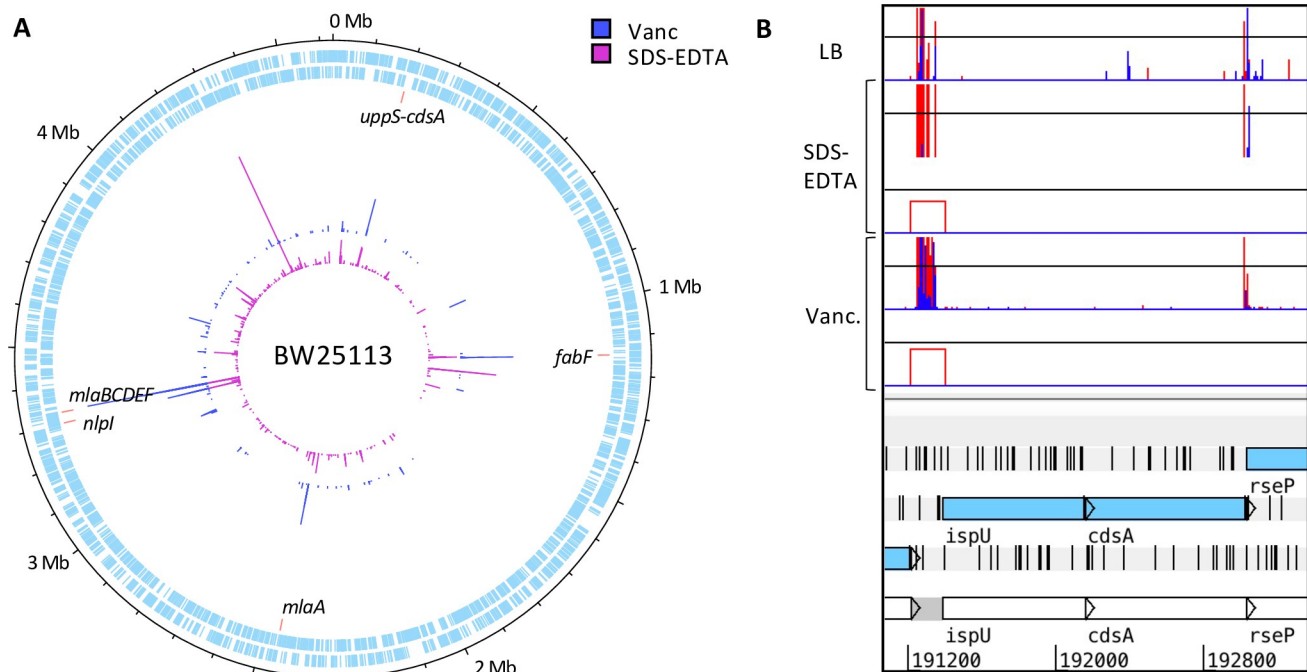

**Fig 8. The *uppS-cdsA* operon.** (A) Genome map showing the location and frequency of transposon insertion events that restore the viability of a *yhcB* mutant under toxic growth conditions: LB supplemented with vancomycin, or SDS and EDTA. (B) TIS data of the *uppS (ispU)* and *cdsA* operon. (Note that *uppS* is annotated as *ispU* in the BW25113 reference genome). Red and blue data correspond with the transposon orientation at the point of insertion; equivalent to sense and anti-sense directionality of the transcriptional read-out from the internal Tn promoter respectively. The height of the bar corresponds with mapped sequencing read frequency, capped at 20 in these images. The red boxes underneath each suppressor dataset represent significant differential abundance of insertions of the condition compared to the control, identified by AlbaTraDIS.

contained either the reported *mlaA** mutation or a variation of this mutation (mut4 and mut6), while one isolate had a genome inversion that resulted in separation of the *mla* operon from its promoter region (mut2). The *mla* inversion mutation restored resistance to both vancomycin and SDS-EDTA, consistent with the earlier TIS data, but did not fully restore cell size (S9 Fig). The *mlaA** mutation is well characterised and is toxic in a wild-type background as it results in a flux of phospholipids into the OM outer leaflet, which triggers PldA-mediated phospholipid degradation and a fatty acid-mediated signalling cascade that results in increased LpxC stability and increased LPS production [95,96]. However, the *mlaA** mutation is restorative in a Δ*yhcB* mutant, restoring both cell size and resistance to vancomycin (S9 Fig). We also identified a single nucleotide polymorphism (SNP) in *lpxC* that restored resistance to vancomycin and SDS-EDTA but did not restore cell size (mut1). These data are suggestive of either a defect in LPS synthesis or export, or an imbalance in LPS to phospholipid at the outer membrane of a Δ*yhcB* mutant.

Finally, two isolates (mut3 and mut5) each had a SNP that resulted in a single amino acid substitution in CdsA. We used Phyre2 to predict the structure of CdsA [97]. The predicted structure was very similar to the experimentally determined structure of CdsA, derived from *Thermotaga maritama*, with an RMSD value of 0.404 Å (S9D Fig) [98]. The active site of CdsA is a conserved, negatively charged pocket occupied by two cation cofactors that are critical for function (S9D Fig). Mutations within this region that hinder cation binding result in decreased enzyme activity [98]. The mutations identified within *cdsA* in our revertant suppressor mutants each result in substitution of a serine residue proximal to the metal ion binding site (S231L and S239G, respectively; S9D Fig). A mutation of S223C in *T. maritama* CdsA, which

corresponds with residue S239 in *E. coli*, results in severely reduced CdsA function [98]. The two mutations within CdsA in the revertant suppressor mutants likely result in decreased function of CdsA. Both mutants with a SNP in CdsA (mut3 and mut5) restored resistance to both vancomycin and SDS and EDTA, in addition to cell size (S9 Fig).

## Characterisation of a *yhcB*-mutant envelope

Our genetic screening revealed a connection with phospholipid biosynthesis (*cdsA*, *fabF*) and trafficking (*mla*). Previous research has shown that fatty acid availability determines the cell size via phospholipid biosynthesis [60]. Increased phospholipid production resulted in a larger cell size that was reminiscent of a Δ*yhcB* mutant [60]. As such, we hypothesised that phospholipid production might be increased in a *yhcB* mutant. We first investigated whether membrane "lipid ruffles" resulting from increased phospholipid synthesis, resembling those reported by Vadia *et al.* [60], could be detected in a *yhcB* mutant. Microscopic evaluation of cells stained with MitoTracker revealed that *E. coli* BW25113 membranes stained uniformly. In contrast, intensely stained spots indicative of accumulation of lipid structures were observed in the *yhcB* mutant (S10A Fig). Subsequent imaging of cells by transmission electron microscopy revealed the presence of internal membrane structures (S11 Fig). These structures were further examined in fast frozen cells after freeze substitution avoiding artefacts associated with chemical fixation [99]. Internal membranes showed a range of different morphologies including tubular and vacuolar structures (Figs 9A and S12) as well as striking tightly-stacked membrane arrays (yellow arrowheads S12 Fig).

Given the increase in lipid biogenesis noted above we next investigated the composition of phospholipids species. We extracted the total phospholipid species using the Bligh and Dyer method [100], and separated these by thin layer chromatography [101]. Following growth in LB, we observed an additional phospholipid species that was present in both strains, but only the Δ*yhcB* mutant strain in stationary phase (Figs 9B and S10B). This was not observed when the mutant was grown in M9-glucose (S10C Fig). By using known lipid standards, the additional spot in the Δ*yhcB* mutant stationary phase was identified as most likely lysophosphatidylethanolamine (LPE; Fig 9B); this was confirmed by targeted mass spectrometry analysis (S13 Fig and S8 Table).

LPE can be derived from PE hydrolysis or lipoprotein maturation. However, in a cell with a stressed envelope, the primary source of LPE accumulation occurs via PE hydrolysis [102]. Two OM proteins, PldA and PagP, maintain the OM asymmetry by cleaving phospholipids that accumulate in the outer leaflet of the OM. PldA degrades phospholipids while PagP, a Lipid A palmitoyltransferase, cleaves an acyl chain from PE and catalyses its transfer to Lipid A resulting in hepta-acylated LPS [103]. Both reactions result in an increase in free LPE. As PagP is usually inactive in the OM and activity is stimulated in response to migration of phospholipids into the outer leaflet [104], measuring the amount of hepta-acylated LPS is an established proxy for detecting loss of OM asymmetry [105]. We quantified the amount of hepta-acylated lipid A relative to hexa-acylated lipid A and identified a significant increase in hepta-acylated lipid A in the Δ*yhcB* mutant compared to the wild-type; a phenotype that is exacerbated by prolonged growth (Figs 9C and S10D). Together these data suggest that excess phospholipid production in a *yhcB* mutant background results in an increase in cell size and loss of OM asymmetry, resulting in hepta-acylation of lipid A and accumulation of LPE. We quantified the lipid species of both membranes by [32P]-radiolabelling and observed a significant (p = 0.018) increase in phospholipids in the cytoplasmic membrane of the *yhcB* mutant relative to the wild-type following prolonged growth in LB (S14 Fig). However, there was no significant difference (p = 0.14) in the ratio of phospholipid:LPS at the OM of the *yhcB* mutant

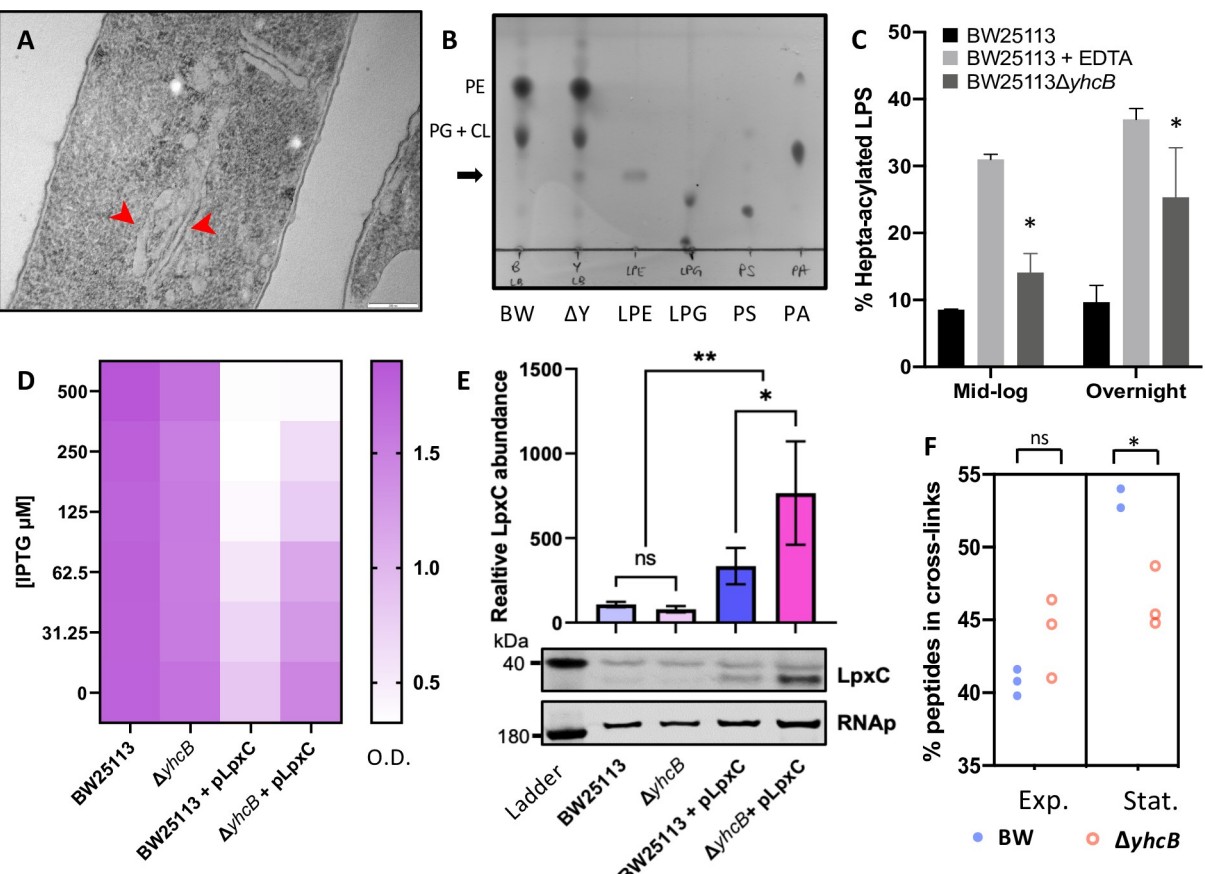

**Fig 9. The *yhcB* mutant contains extra membrane and has an altered cell envelope.** (A) Representative transmission electron micrograph (TEM) image of a Δ*yhcB* mutant cell processed without primary fixation by fast freezing and freeze substitution. Internal membrane structures indicated by red arrows. Scale bar = 200 nm.(B) Phospholipid species extracted from BW25113 (BW) and the Δ*yhcB* mutant (ΔY) grown in LB broth until stationary phase and separated by thin layer chromatography (TLC) using a chloroform:methanol:water (65:25:4) solvent system. Standards of lysophosphatidylethanolamine (LPE), lysophosphatidylglycerol (LPG), phosphatidylserine (PS) and phosphatidic acid (PA) were loaded alongside for comparison. The additional spot in the ΔY sample is indicated with an arrow. (C) Analysis of [$^{32}$P]-labeled lipid A extracted from mid-logarithmic growth and overnight growth cultures of BW25113 and Δ*yhcB* strains grown in LB, presented as %hepta-acylated lipid A. As a positive control for lipid A palmitoylation, BW25113 cells were treated with 25 mM EDTA for 10 min prior to extraction. The mean percentage of hepta-acylated lipid A and standard deviation were calculated from samples prepared in triplicate. Student's *t*-tests: *$P < 0.05$ compared with BW25113. (D) Optical Density (OD) measurements of BW25113 and a Δ*yhcB* mutant +/- pLpxC taken after 16 h growth in LB with and without IPTG induction of *lpxC*. (E) Relative LpxC abundance in BW25113 and a Δ*yhcB* mutant +/- pLpxC after 16 h growth in LB supplemented with 31.25 μM IPTG, measured by western blot analysis and normalised to RNA polymerase. Statistical significance ($P < 0.05$) determined using a one way ANOVA with Tukey's correction for multiple comparisons, with * $P ≤ 0.05$ and ** $P ≤ 0.01$. (F) Comparison of the percentage of peptides in peptidoglycan crosslinks between strains at two stages of growth. Unpaired t-test with Welch's correction: *$P < 0.05$. Exponential phase cells were collected at an OD of 0.4, Stationary phase cells were harvested after 16 h at an OD of ~4.0. Abbreviations: phosphatidylethanolamine (PE); phosphatidylglycerol (PG); cardiolipin (CL); exponential phase (Exp.); stationary phase (Stat.).

compared to the wild-type (S14 Fig). We next examined LpxC abundance in the Δ*yhcB* mutant via western blotting. During the experiment we observed that the Δ*yhcB* mutant is more tolerant of *lpxC* overexpression (Fig 9D). There was no detectable difference in the amount of LpxC between the wild-type and Δ*yhcB* mutant, however in the pLpxC positive controls we observed a significant increase in LpxC abundance in the Δ*yhcB*/pLpxC strain compared to the BW25113/pLpxC strain following induction with 31.25 μM IPTG (Fig 9E). As induction was comparable, this suggests that LpxC may be more stable in a Δ*yhcB* mutant and altered proteolytic regulation.

Cell membrane synthesis must be coordinated with peptidoglycan biogenesis to maintain cell envelope integrity. As the peptidoglycan layer defines cellular morphology, and we had observed a swollen morphology of *yhcB* mutants, we hypothesised that the peptidoglycan layer of a *yhcB* mutant may be compromised. Indeed, our genetic evidence supports the hypothesis that mutations (*dapF*, *dacA*) that weaken integrity of the peptidoglycan are lethal. We therefore analysed the muropeptide composition of the cell wall. Cells were grown in LB and collected at both exponential and stationary phase. Peptidoglycan was purified and digested by the muramidase cellosyl, and the resulting muropeptides were analysed by HPLC [106]. There were no significant differences in the muropeptide species identified (S15 Fig and S9 Table). However, we did identify differences in the amount of crosslinking between strains and growth phases (Fig 9E). While the degree of peptidoglycan crosslinking in the wild-type strain was increased in stationary phase compared to exponential phase, consistent with the literature [77], this same transition was not observed in the *yhcB* mutant, suggesting either a limitation in the rate of peptidoglycan crosslinking or an increased rate of crosslinking hydrolysis in a *yhcB* mutant. The muropeptide analysis, together with the phospholipid analysis, support our finding that a *yhcB* mutant is unable to sustain rapid exponential growth and indicate a defect in the ability to transition into stationary phase.

### The role of the *uppS-cdsA* operon in a *yhcB* mutant

In addition to a synthetically lethal relationship with genes involved in peptidoglycan assembly, recycling and remodelling (*mepS*, *mepM*, *ldcA*, *nlpI*) our TIS data also indicated that mutations in the ECA biosynthetic pathway that sequester undecaprenol phosphate (Und-P) are lethal in a *yhcB* mutant. We hypothesised that the availability of Und-P might be limited in a Δ*yhcB* mutant, affecting peptidoglycan synthesis [80,107,108]. One mechanism to suppress Und-P limitation is to upregulate expression of *uppS*, the gene required for Und-P synthesis. Indeed, our TIS data reveal insertions that suppress the *yhcB* phenotype in the promoter region to *uppS* (Fig 8B). Complementation of a *yhcB* mutant with ectopically expressed *uppS*, but not *cdsA*, suppressed both the vancomycin and SDS-EDTA sensitivity defect of a *yhcB* mutant (Fig 10A).

During our analysis we noted that *uppS* and *cdsA* are highly co-conserved across the bacterial kingdom (S16 Fig) and share a conserved operon structure in many bacterial phyla (S17 Fig) [109]. We hypothesise that the upregulation of *uppS* would result in increased expression of *cdsA*. Evaluation by reverse transcription (RT)-PCR demonstrated an approximate 2-fold increase in *uppS* and *cdsA* expression in a *yhcB* mutant compared to wild-type (Fig 10B). Meanwhile ectopic expression of an RNA antisense to *cdsA*, designed to knock down *cdsA* expression, restored tolerance to SDS-EDTA but not vancomycin (Fig 10C). The sensitivity to vancomycin is likely due to polar effects of the antisense construct on expression of upstream *uppS* as the loss-of-function mutations we identified above in CdsA conferred resistance to vancomycin in a *yhcB* mutant (S9A Fig). Microscopic evaluation of these strains found that *cdsA* knockdown also restored the cell dimensions of a *yhcB* mutant grown in LB (S18 Fig), consistent with S9C Fig. Together, these data suggest loss of *yhcB* results in an increased demand for the Und-PP synthase, resulting in upregulation of the *uppS-cdsA* operon. Increased expression of *cdsA* in turn gives rise to excess phospholipid synthesis, resulting in detergent sensitivity and enlarged cells.

## Discussion

Here we have applied a high-density mutagenesis screen to comprehensively map the genetic interactions of a single gene, *yhcB*. With thousands of deleterious and restorative genetic

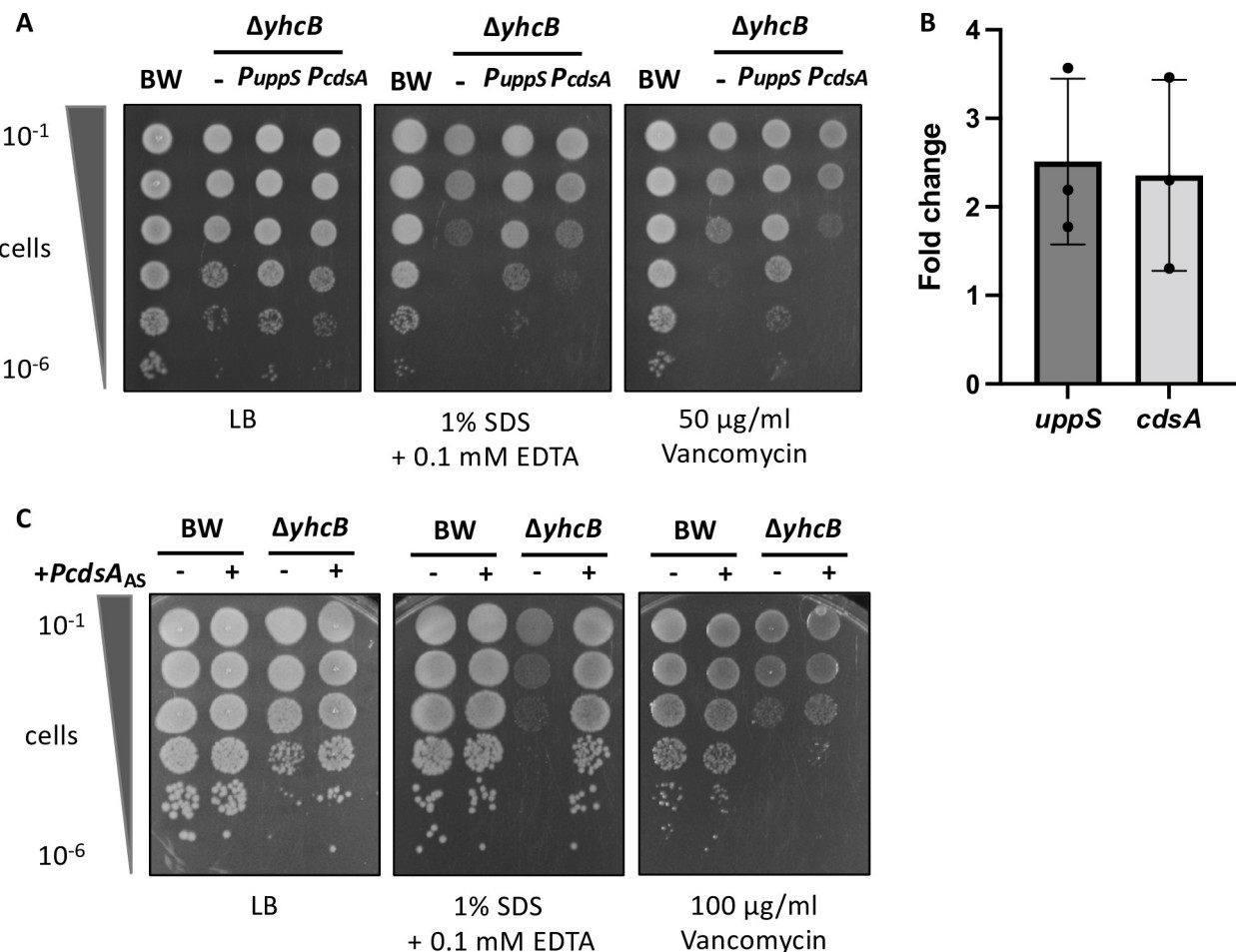

**Fig 10. Expression of the *uppS*-*cdsA* operon.** (A) Cultures of BW25113; Δ*yhcB*; Δ*yhcB*/P*uppS* and Δ*yhcB*/P*cdsA* strains grown in LB, normalised to an OD$_{600}$ of 1.00 and 10-fold serially diluted before inoculating LB agar plates supplemented with vancomycin or SDS and EDTA. (B) RT-PCR data presented as the relative fold change in abundance of *uppS* and *cdsA* transcripts in a *yhcB* mutant compared to the BW25113 control. Data are normalised to *gyrA* prior to comparative analyses. Data were collected for three biological replicates. The experiment was repeated twice, a representative plot is shown. (C) Cultures of BW25113; BW25113/P*cdsA*$_{AS}$; Δ*yhcB* and Δ*yhcB*/P*cdsA*$_{AS}$ strains grown in LB for 5 h, normalised to an OD$_{600}$ of 0.05 and 10-fold serially diluted before inoculating LB agar plates supplemented with vancomycin or SDS and EDTA. The P*cdsA*$_{AS}$ construct contains the antisense nucleotide sequence of *cdsA*. Expression was not induced, leaky expression was sufficient for complementation purposes.

interactions discovered, to the best of our knowledge no other gene has been scrutinised genetically at such scale and resolution, demonstrating that TIS is a powerful tool that can transform our understanding of poorly characterised genes. Our data reveal that a Δ*yhcB* mutant has a cell envelope permeability defect, dysregulated cell size, and epistatic interactions with multiple pathways of cell envelope biogenesis. The defect introduced by deletion of *yhcB* is most severe following rapid growth in rich media, when the demand for synthesis of cell envelope material is greatest. Toxicity progresses with sustained growth in rich medium, with a decrease in cell viability observed upon transition to stationary phase, a period of growth when *de novo* synthesis machineries are switched off. Taken together, these observations suggest that a *yhcB* mutant is either limited in the ability to recycle existing components of the envelope during synthesis, or limited in *de novo* synthesis of one or more cell envelope components.

The pathways for synthesis of the different components of the Gram-negative cell envelope are intrinsically linked: they share several common precursors and tightly interconnected

feedback mechanisms (Fig 11A). If the rate of peptidoglycan biosynthesis is decreased, the rate of LPS biosynthesis needs to be decreased accordingly to avoid the toxic effects of LPS accumulation [95,110]. Moreover, a recent paper identified that deletion of the *mla* pathway confers a fitness advantage in cells exposed to Fosfomycin, a MurA inhibitor, connecting decreased PG synthesis with a requirement for decreased phospholipid trafficking [111]. Similarly, when LptC, a component of the LPS export system, is depleted the relative abundance of proteins involved in phospholipid biosynthesis and trafficking are also depleted indicating a feedback mechanism that couples phospholipid synthesis and export with LPS demand [112]. In a *yhcB* mutant we observed weakened peptidoglycan integrity and a loss of OM asymmetry, coupled with an increase in cell size and internal phospholipid membrane structures (Fig 11B). These findings connect YhcB with every layer of the cell envelope.

Our observations of excess phospholipid membranes and an increased cell size in a *yhcB* mutant resemble the phenotype reported by Vadia *et al.* [60] that results from increased phospholipid synthesis. We hypothesise that the increase in phospholipid production results in loss of OM asymmetry, with phospholipids integrating into the outer leaflet of the OM compromising the barrier function of the cell envelope. This hypothesis is supported by several lines of evidence. First, phase contrast microscopy with MitoTracker labelling and electron micrographs revealed internal lipid structures in the *yhcB* mutant, while quantification by radiolabelling showed a >2-fold increase in phospholipids that partitioned with the cytoplasmic membrane under a sucrose gradient. Second, inhibition of phospholipid synthesis by knockdown of CdsA, amino acid substitutions proximal to the metal ion binding site of CdsA, or loss of FabF, which regulates membrane fluidity, are restorative to a *yhcB* mutant. Furthermore, treatment with sub-inhibitory concentrations of cerulenin, an antibiotic that targets the FabB and FabF elongation step of fatty acid synthesis, is beneficial to growth of a *yhcB* mutant [113]. Third, a *mlaA** mutation, which increases mislocalisation of phospholipids at the OM triggering subsequent phospholipid degradation by PldA and is toxic in a wild-type background, is restorative in a *yhcB* mutant: it restores cell size and re-establishes the barrier function of the OM. Fourth, an increase in LPS production through the loss of the essential protease FtsH can be tolerated or the introduction of *mlaA** which upregulates LPS production is restorative in a *yhcB* mutant. Overall, mutations that result in a decrease of phospholipid biosynthesis and trafficking, or an increase in LPS biosynthesis are restorative. Although we found no significant decrease in the ratio of LPS:PL at the OM compared to wild-type, one hypothesis is that the relative balance can be maintained through active degradation of accumulating phospholipids as observed in our PagP assay.

We found that mutations that weaken PG integrity are lethal in a *yhcB* mutant indicating that the rate of PG synthesis is also not coordinated with cell expansion. While the composition of the PG in a *yhcB* mutant resembles that of the wild-type, our data reveal that loss of PG synthesis by PBP1A in the elongasome is restorative. In contrast, loss of PG synthesis by PBP1B in the divisome had a negative effect while mutations that stimulate PG synthesis via the divisome are restorative. Given the location of YhcB within the IM, its interaction with the elongasome, and the phenotypes of a deletion mutant, an attractive hypothesis is that YhcB is involved in switching PG synthesis between the elongasome and divisome. In support of this hypothesis a recent paper reported the interaction and co-purification of YhcB with proteins of the divisome [42].

Another attractive hypothesis is that YhcB has a role in sensing the availability of Und-P, or recycling/synthesis of Und-P. This latter hypothesis is supported by several observations. First, mutations in the ECA biosynthesis pathway that sequester Und-P in the form of Und-P-linked intermediates are lethal. Second, a *yhcB* mutant is reported to be more sensitive to bacitracin [113], which targets BacA: an enzyme responsible for recycling approximately three quarters

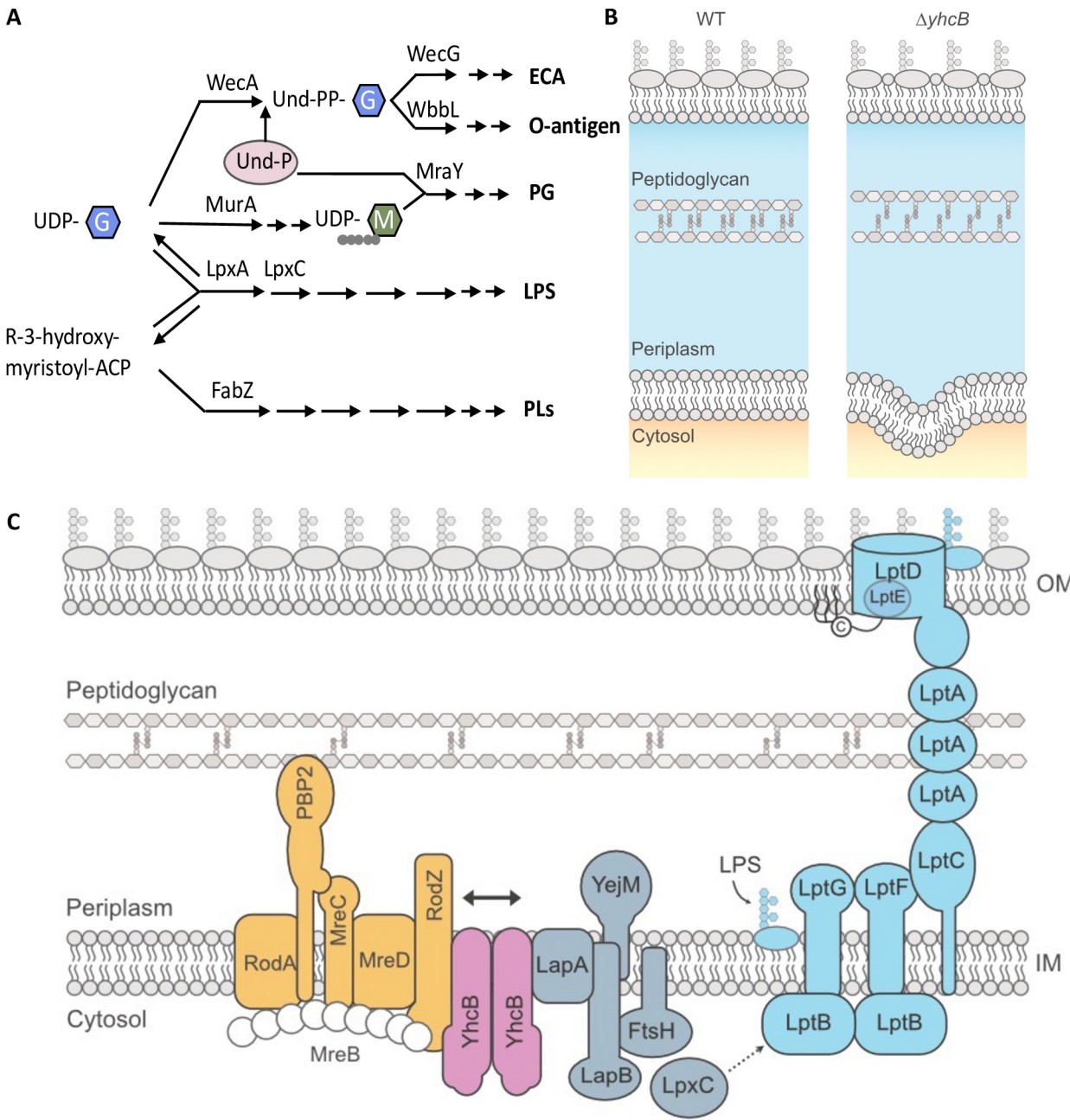

**Fig 11. The pathways of cell envelope biosynthesis and relationship with YhcB.** (A) Schematic overview of the relationships between the biosynthetic pathways of individual components of the cell envelope. (B) A schematic of the imbalanced cell envelope of a *yhcB* mutant in comparison to wild-type (WT). The *yhcB* mutant suffers from defects in OM asymmetry, excessive phospholipid synthesis and altered crosslinking in the peptidoglycan layer. (C) The positional context of YhcB in conjunction with the reported protein interaction partners of YhcB *in situ*. YhcB interacts with both the elongasome (amber; depicted here as mediated via RodZ, but YhcB is reported to interact with RodA, MreD and MreC in addition to RodZ) and LapA from the complex that regulates LpxC degradation (grey). YhcB is spatially positioned to coordinate peptidoglycan expansion with LPS export, represented by the solid black double arrow. For simplicity, the LPS biosynthesis pathway has been replaced by a dashed arrow from LpxC, which exacts the first committed step of LPS synthesis, to the Lpt bridge, which facilitates LPS export. Abbreviations: Enterobacterial Common Antigen (ECA); Lipopolysaccharide (LPS); Peptidoglycan (PG); Phospholipid (PL); Undecaprenyl phosphate (Und-P); Undecaprenyl diphosphate (Und-PP); Uridine diphosphate (UDP); N-acetylglucosamine (G); N-acetylmuramic acid (M); Acyl Carrier Protein (ACP). Interaction of YhcB with RodZ and LapA has been reported by Li *et al.* [37] and Hu *et al.* [116].

of Und-PP back to Und-P following release of its substrate in the periplasm. Third, *mrcB* (encoding PBP1B) was identified in the list of genes in which an epistatic interaction with *yhcB* causes a fitness defect (S4 Table), consistent with data that deletion of *mrcB* in cells with low levels of Und-P results in heterogenous cell lysis [114]. Fourth, upregulation of *uppS*, but not *cdsA*, was universally restorative in a *yhcB* mutant. Lastly, a recent paper quantifying available Und-P in wild-type cells and *wec* mutants found that disruptions to the ECA pathway resulted in an increase in Und-P [115]. One explanation put forward by the authors is that a feedback mechanism may exist to regulate Und-P synthesis in response to sequestering Und-P-linked intermediates. We observed an increase in transcription of the *uppS-cdsA* locus in the ΔyhcB mutant compared to the wild-type; as such, upregulation of this locus may also be indicative of disruption to Und-P turnover.

The observation that the *uppS-cdsA* operon structure is conserved across bacteria reveals the tight linkage between phospholipid production and the synthesis of peptidoglycan. The observation that cell size, resistance to vancomycin, and resistance to SDS-EDTA can be decoupled in suppressor screens indicate that YhcB interacts with multiple cell envelope biogenesis pathways. Indeed, in addition to forming a dimer and reports of larger homo-multimer structures [37,39,42], YhcB is known to interact with several components of the elongasome [37], and has also been identified as an interaction partner of LapA [37,116]. LapA (YciS) forms a complex with LapB (YciM), FtsH and YejM to regulate LpxC stability and thus LPS abundance [38,117,118]. The role of LapA is unclear but deletion of *lapA* results in minor accumulation of incomplete LPS precursors [33]. The interaction of YhcB with the elongasome and the LpxC regulation complex via LapA positions YhcB at the interface between two of the major complexes that coordinate envelope synthesis (Fig 11C). This hypothesis is supported by the observation that deletion of different YhcB domains gives rise to different chemical sensitivity profiles.

In conclusion, YhcB plays an important regulatory function at the interface between the cell envelope biogenesis pathways, and this function is mediated by physical interactions between members of the elongasome and LPS regulation systems. While the precise molecular events facilitating the regulation of these pathways has yet to be elucidated, as *yhcB* is conserved among Gammaproteobacteria, is required for tolerance to polymyxin B, colistin, and vancomycin, and is necessary for the colonisation of different hosts [51,119,120], it provides a novel antimicrobial target for exploitation against clinically important pathogens.

## Materials and methods

### Strains, media and growth conditions

The parent strain for this work was *E. coli* K-12 strain BW25113. Gene deletion mutants were constructed by P1 transduction and using the Keio library strains as donors [121,122]. The gene *ftsH* was deleted by amplification of the kanamycin resistance cassette using pKD4 as a template, and following the Datsenko-Wanner protocol for single-step gene inactivation [69]. For the *yhcB* mutant, the kanamycin cassette was removed using the pCP20 vector [69]. The mutant was confirmed by PCR and Sanger sequencing followed by whole genome sequencing. Our strain did not carry the SNPs identified in either of the Keio *yhcB* mutants (ENA accession: ERP127382). For a full list of strains used in this study see S11 Table. Bacteria were grown in Lysogeny Broth (LB, Miller recipe: 10 g tryptone, 5 g yeast extract, 10 g NaCl) medium or on LB plates (LB supplemented with 1.5% nutrient agar) and incubated at 37°C. When required, media were supplemented with 50 μg/ml kanamycin or 100 μg/ml carbenicillin. The M9-glucose recipe used was:1x M9 salts (Sigma Aldrich), 200 μl filter sterilised 1 M MgSO₄, 10 μl filter sterilised 1 M CaCl₂, and 2 ml of either a 20% (w/v) D-glucose or 20% (v/v)

glycerol solution per 100 ml. For micro-dilution spot plates, unless otherwise stated, bacteria were grown overnight in 5 ml LB medium at 37˚C with aeration. Cultures were normalised by optical density to an $OD_{600}$ of 1.00, 10-fold serially diluted in LB, and 2 µl of each dilution was inoculated onto LB agar plates.

## Complementation assays

The complementation pBAD-*yhcB* vector was constructed by restriction-free cloning using BW25113 genomic DNA as a template for the *yhcB* gene. The BW25113 and *yhcB* strains were transformed with a pBAD-Myc-His-A plasmid with and without the *yhcB* coding sequence under the control of the arabinose promoter. The truncated YhcB vectors were constructed via a blunt-end cloning method using the pBAD-*yhcB* vector as a template and PCR amplification of the larger backbone fragments, which were then treated with T4 PNK followed by a ligation step, using NEB products and following user instructions. Additional copies of *cdsA* and *uppS* were cloned, independently, between the XbaI and HindIII restriction sites of a pASK-IBA2C plasmid under the control of an anhydrotetracycline-inducible promoter. Antisense *cdsA* was cloned into a pACYCDuet-1 vector via Gibson assembly using the NEB Gibson assembly master mix. See S12 and S13 Tables for a full list of plasmids and oligonucleotides, respectively, used in this study.

## Polymyxin B TIS screen

An amended version of the Andrews broth microdilution protocol in 96-well plate format, adjusting the starting inoculum to an initial $OD_{600}$ of 0.05 and the growth medium to LB, was used to identify an initial inhibitory concentration range or polymyxin B [123]. Growth curve experiments of *E. coli* BW25113 were then repeated in 50 ml LB supplemented with polymyxin B in glass flasks, reflecting the conditions of the TIS screen. A concentration of 0.2 µg/ml polymyxin B was identified as sub-inhibitory for growth of BW25113.The transposon library was inoculated into 50 ml LB medium with and without 0.2 µg/ml polymyxin B (Sigma Aldrich), in duplicate, at a starting optical density of 0.05, equivalent to a copy number of ~2,500 of each mutant, and grown to $OD_{600}$ = 1.00, at 37˚C with aeration. Cells were harvested, and genomic DNA prepared for sequencing. As we knew the density of this library [44], we estimated the number of mapped reads, and therefore sequencing coverage, needed to ensure sufficient sampling of the library using the equation $I = s - s\left(\frac{s-1}{s}\right)^{n}$, where I = insertions, and n = number of mapped reads (S1B Fig). For a theoretical library of 1 million mutants, assuming no loss of mutants and an equal chance of each transposon junction being sampled, ~2.3M reads are needed to ensure sampling of 90% of the library, with diminishing returns with further sequencing. Obtaining >4.2 M reads should enable sampling of 99% of the possible unique insertion sites. Therefore, we collected >2 M sequencing reads per replicate, and a combined total of >4.2 M reads per condition, to give us confidence that any observed loss of mutants is not due to insufficient sampling.

## Microscopy

Samples were taken directly from overnight culturesgrown at 37˚C in their respective media for 16 h and diluted to an $OD_{600}$ of 0.10. 5 µl cells were spread on a 1 mm glass slide pre-treated with 5 µl poly-L-lysine (Sigma Aldrich). A Nikon 90i eclipse microscope was used to capture differential interference contrast (DIC) images of cells, using a 40x objective lens with a Nikon immersion oil. These images were collected at the University of Birmingham. Cell dimensions were measured using ImageJ [124]. Fluorescent labelling of lipids was previously described

[60]. These experiments were done at the University of Queensland. M9-glucose overnight cultures were diluted into fresh LB medium and after 2 h of growth at 37˚C, 0.2 μM Mito-Tracker Green FM (Invitrogen) was added to each sample to label lipids. After incubation for a further 1 h, bacteria from 1 ml samples were harvested by centrifugation and resuspended in 500 μL of FM4-64FX (Invitrogen) at 5 μg/mL, to label *E. coli* membranes, and incubated for 5 min at room temperature. Bacteria were embedded on 1% agarose pads and imaged using a confocal microscope Inverted LSM 880 Fast Airyscan (63x/1.40 OIL). Electron microscopy was performed at the University of Queensland. Conventional fixation and embedding in resin was undertaken using a protocol adapted from Fassel *et al.* [125]. Briefly, bacteria were applied to dishes with glass coverslips coated in poly-L-lysine and then fixed in 2% PFA, 2.5% Glutaraldehyde, and 0.075% Ruthenium Red. Samples were then washed 3x 10 min in PBS and postfixed in 1% osmium, then again in 1% osmium with 3% potassium ferricyanide. Bacteria underwent serial dehydration in increasing concentrations of ethanol and were then infiltrated and embedded with Epon resin. Fast freezing and freeze substitution of bacteria was undertaken as described previously [99,126]. Briefly, bacteria were applied to carbon-coated poly-L-lysine treated sapphire discs and frozen in a Leica EMPACT 2 high pressure freezer. They were then freeze substituted in a EM AFS 2 system (Leica Microsystems GmbH, Wetzlar, Germany) and embedded into Epon resin. Ultra-thin sections were obtained using a Leica Ultracut UC6 Ultramicrotome and micrographs acquired on a JEOL 1011 transmission electron microscope equipped with a Morada CCD camera.

## Phospholipid extraction

Total phospholipids were extracted using an amended version of the Bligh-Dyer method [100]. 10 ml of culture at an $OD_{600}$ ~ 4.00, or 100 ml of culture at an $OD_{600}$ of 0.40 was centrifuged at 4˚C to harvest cells. The supernatant was discarded, and pellet resuspended in 1 ml $ddH_2O$ and transferred to a glass tube. 1.25 ml chloroform and 2.5 ml methanol were added to the sample using glass pipettes. The sample was vortexed for 20 s to create a single-phase solution which was then incubated at 50˚C for 30 min. A further 1.25 ml chloroform and 1.25 ml water was added to the sample to create a 2-phase solution and incubated again at 50˚C for 30 min. The sample was centrifuged at 400 x $g$ for 10 min at RT and the lower organic phase containing phospholipids was transferred to a new glass tube using a glass Pasteur pipette. Chloroform was evaporated by placing the glass tubes in a heat block at 50˚C under a stream of nitrogen.

## Thin layer chromatography

10 μl of sample was spotted onto the origin of a TLC silica gel membrane (Merck) using a 5 μl glass capillary tube (Sigma Aldrich). Once dry, the membrane was transferred to an equilibrated solvent system of 65:25:4chloroform:methanol:water [101]. The samples were separated until the solvent front had migrated sufficiently from the origin then the membrane was removed from the solvent tank and was air dried at RT. Samples were stained with phosphomolybdic acid (PMA) 10% solution in ethanol and heated with a heat gun to activate the PMA until lipid species were visible. Lipid standards were purchased from Avanti Polar Lipids and handled according to manufacturer instructions.

## Lipid A palmitoylation assay

Lipid A labelling, extraction and analysis were done as described and demonstrated previously [105]. As a positive control, BW25113 was exposed to 25 mM EDTA for 10 min to induce PagP mediated palmitoylation of Lipid A, before harvesting cells by centrifugation.

## Membrane lipid composition analyses

Experiments to determine the phospholipid distribution between IM and OM and the ratio of phospholipid to LPS at the OM were repeated exactly as described previously [127], with one addition, cell cultures were harvested at stationary phase (18 h growth) in addition to mid-exponential phase of growth ($OD_{600}$ = 0.5). Briefly, cultures were grown in triplicate in either 5 ml (stationary) or 10 ml LB (mid-exponential) with 1 μCi/ml [$^{32}$P]-disodiumphosphate (Perkin Elmer, NEX011001MC) to label phospholipids and LPS. Cells were harvested by centrifugation, washed, and resuspended in 5 ml of 20% sucrose in 10mM Tris-HCl (pH 8.0 (w/w) containing 1 mM PMSF and 50 μg/ml DNase I), and lysed using a high-pressure French press. The cell lysate was centrifuged and the supernatant subjected to sucrose density fractionation over 16 h, as described previously [127]. IM (7–9) and OM (12–14) fractions were collected and concentrated into 320 μl TBS and LPS and/or phospholipid extracted and dried as described previously [128]. Dried lipids were resuspended in 50 μl solvent (2:1 chloroform:methanol for phospholipids, 1% SDS for LPS), and equal volumes were mixed with Ultima Gold scintillation fluid (Perkin Elmer). [$^{32}$P]-counts were measured using scintillation counting (MicroBeta$^2$, PerkinElmer). For each strain, scintillation counts of OM phospholipids were divided by the IM phospholipids to get the OM:IM ratio; while scintillation counts of OM LPS were divided by the counts of OM phospholipids to obtain the LPS/PL ratio. Independently, separation of IM and OM fractions in samples following overnight growth was confirmed by growing cell cultures overnight in LB with 1 μCi/ml [$^3$H]-glycerol. Cells were harvested and membranes separated as described above. Following separation by centrifugation, 15 fractions were collected from the top of each tube and scintillation counts recorded for 350 μl of fractions 3–14.

## Peptidoglycan extraction and analysis

Cell cultures were grown in LB medium at 37°C and harvested at an OD of 0.4 and 4.00, in triplicate. After pelleting the cells and resuspending in ice-cold water, the cell suspension was dropped into 8% boiling SDS solution. Peptidoglycan was purified, and muropeptides were released with cellosyl and analysed by HPLC as described [106].

## Detection of LpxC

BW25113 and the isogenic *yhcB* mutant were transformed with the *lpxC* over-expression vector pTrc99a-*lpxC* (pLpxC). Strains were inoculated in triplicate into a 96 well plate with or without the addition of IPTG at a concentration range (500–32.25 μM). Strains were incubated at 37°C for 16 h in a Polarstar plate reader and the $OD_{600nm}$ measured every 30 min. After growth, bacterial cells incubated with 31.25 μM IPTG were centrifuged and resuspended in Laemmli sample buffer (Sigma Aldrich) and normalised by wet weight of the cell pellet. Samples were boiled for 5 min and briefly centrifuged prior to loading onto NuPAGE 4–12% bis tris SDS-PAGE gels (Invitrogen). Proteins were transferred to nitrocellulose membranes and blocked for 30 min in 5% milk at room temperature. RNA polymerase and LpxC specific antibodies were added at a 1:5,000 dilution and were incubated with the membranes overnight at 4°C. Membranes were washed four times in TBST (tris buffered saline 0.1% Tween20) prior to the addition of either anti mouse (IRDye 680LT goat anti-mouse IgG) or anti rabbit (IRDye 800CW goat anti-rabbit IgG) secondary antibodies at a 1:10,000 dilution for 1 h at room temperature. Membranes were washed five times in TBST prior to capturing the fluorescent image using a Chemidocimager (Bio-rad). Western blot band abundancies were measured using the OdessyCLx imaging system. LpxC band intensities were normalised to the average intensity of the corresponding RNA polymerase band. The normalised data were analysed using Graphpad

Prism (v 9.0.0) and statistical significance ($p < 0.05$) determined using a one way ANOVA with Tukey's correction for multiple comparisons, with $^*$ $p \leq 0.05$ and $^{**}$ $p \leq 0.01$.

## Mass spectrometry analysis

The unknown spot was isolated by first separating lipid species by TLC as above, excising the unstained region from the silica membrane and resuspending the lipid species in chloroform. An identical sample was processed in parallel, with PMA staining, to use as a guide to identify the spot position. For control, the plate with excised membrane was then treated with PMA to develop the lipid spots and confirm excision of the correct species. The extracts were analyzed by LC-MS/MS on a Shimadzu Nexerau HPLC (Japan) coupled to a Triple Tof 5600 mass spectrometer (ABSCIEX, Canada) equipped with a duo electrospray ion source. 5 μl of each extract was injected onto a 2.1mm x 150 mm Waters, XBridge BEH300 C18, 3.5 μm C18 column (Waters, Ireland) at 200 μl/min and 50C. Linear gradients of 1–40% solvent B over 15 min at 200 μL/min flow rate, followed by 40% to 80% solvent B in 21 min, then 80% to 98% B in 1 min were used for lipid elution. Solvent B was held at 98% for 5 min for washing the column and returned to 1% solvent B for equilibration prior to the next sample injection. Mobile phase solvent A consisted of 60/40 acetonitrile/0.1% formic acid (aq) and solvent B contained 90/9/1 propan-2-ol/acetonitrile/0.1% formic acid (aq). The ionspray voltage was set to 5500 V, for positive detection and -4500 V for negative ion detection, declustering potential (DP) 100V, curtain gas flow 25, nebulizer gas 1 (GS1) 50, GS2 to 60, interface heater at 150˚C and the turbo heater to 500˚C. The mass spectrometer acquired 250 ms full scan TOF-MS data followed by up to 20, 250 ms full scan product ion data in an Information Dependant Acquisition, IDA, mode. Full scan TOFMS data was acquired over the mass range 300–1500 and for product ion ms/ms 80–1800. The 20 most intense precursor ions detected in the TOF-MS scan exceeding a threshold of 120 counts and a charge state of +/-1 to +/-2, positive ion/negative ion detection, were collated for the acquisition of product ion, ms/ms spectra. The data was acquired and processed using Analyst TF 1.6 software (SCIEX, Canada). Lipids were identified using both Analyst 1.6 and LipidView 2.0 software (SCIEX, Canada).

## RT-PCR

Cell cultures were grown overnight then sub-cultured 1:100 into fresh LB and grown for 6 h at 37˚C. Cells were harvested, and RNA was isolated using Trizol and chloroform. RNA was converted into cDNA using a Superscript IV RT-PCR kit following the manufacturer's instructions. DyNAmo HS SYBR Green qPCR Kit was used for quantification of gene expression. The gene *gyrA* was used as an internal control for normalisation of transcript abundance.

## Construction of transposon-mutant libraries

10 ml of 2x TY broth was inoculated with a single colony and grown overnight at 37˚C with aeration. The 10 ml overnight culture was used to inoculate 800 ml 2x TY broth in a 2 L flask and grown at 37˚C with aeration until $OD_{600}$ 0.6–0.9. At the desired OD, cells were collected and stored on ice for 30 min before centrifugation to pellet the cells. Electrocompetent cells were prepared by repeatedly centrifuging cells and resuspending in decreasing amounts of ice-cold 10% (v/v) glycerol. The final resuspension was in 1 ml of 10% glycerol resulting in a dense ~2 ml cell culture. Aliquots of 200 μl cells were distributed between 1.5 ml microcentrifuge tubes. 0.2 μl EZ-Tn5 transposome (Epibio) was mixed with each aliquot of cells and incubated on ice for 30 min. Samples were transferred to pre-chilled 2 mm gap electroporation cuvettes (Cell Projects Ltd.). Cells were pulsed at 2200 V and 2 ml of pre-warmed SOC medium was immediately added to the sample for recovery. Samples were transferred to a 15 ml falcon tube

to allow for maximum aeration and were incubated at 37°C for 2 h. 5 ml of LB broth was added to each 15 ml falcon tube. ~4–5 drops of cells (equivalent to ~200 μl) were spread per LB agar plate supplemented with 50 μg/ml kanamycin. Sufficient cells were inoculated per plate to form non-touching single colonies. Plates were incubated overnight at 37°C for 18 h. Following incubation, 500 μl 30% glycerol-LB broth was added to each plate and using a 'hockey-stick' spreader, colonies were scraped off the surface of the agar plate and pooled. Cells were mixed thoroughly before storing at -80°C.

## Screening of the *yhcB* transposon-mutant library to identify suppressor mutations

Cell cultures were resuspended in 1 ml of $OD_{600} = 1.00$, 200 μl of culture was plated across 5x LB agar plates with or without additional chemical stresses. Assuming a library of ~500,000 mutants (as judged from the unique insertions), this number of cells equates to approximately 2,000 independent copies of each individual transposon mutant. We chose a plate-based screening method as this minimises competition between mutants, allowing for identification of slow growing mutants in addition to strong suppressor mutations. Cultures were diluted to such quantities that single colonies of transposon mutants could be isolated on the plates supplemented with stresses to prevent the growth of satellite colonies and therefore false positive results. Plates were incubated at 37°C for 24 h. Colonies were scraped and pooled for sequencing.

## TIS sequencing

Genomic DNA (gDNA) was extracted and quantified using Qubit dsDNA HS Assay kit (Invitrogen). 1 μg of gDNA was fragmented by mechanical shearing using a bioruptor (Diagenode) using the shearing profile 30 s ON, 90 s OFF at low intensity, resulting in DNA fragments with an average length of ~300 bp. Fragmented DNA was end-repaired using the NEBNext Ultra I kit (New England Biolabs). The fragments were then ligated with an adapter, and the sample was purified in a size-selection step using AMPure XP SPRI beads (Beckman Coulter) before transposon-junctions were enriched by PCR with primers specific for the transposon and the adapter. The Transposon-gDNA junctions were prepared for sequencing by PCR addition of Illumina adapters using the NEBNext Multiplex Oligos for Illumina (New England Biolabs). However, the forward Universal primer was replaced with custom primers that include the Universal primer sequence followed by a 6–9 nucleotide barcode (to introduce complexity and stagger the start of the transposon sequence) and a 22 nt sequence with homology to the transposon at the 3′ end of the primer. Samples were purified after each PCR step using SPRI beads at a ratio of 0.9:1 beads to sample. Finally, the sample was quantified using the KAPA Library Quant Kit (Illumina) Universal qPCR Mix (Kapa Biosystems). Samples were pooled, denatured and diluted to 18 pM and sequenced using an Illumina MiSeq, with 5% (v/v) 20 pMPhiX (Illumina), using 150 cycle v3 cartridges. Data are available at the European Nucleotide Archive (accession: PRJEB43420).

## TIS analysis

Data were first demultiplexed using the Fastx barcode splitter, to remove the 5′ end barcode (inline index) unique to each sample [129]. The BioTraDIS analysis package (version 1.4.5) was used for the remaining data processing [49]. We allow for up to 4 bp mismatches in the transposon pattern matching step. When successfully identified transposon sequences have been identified, the transposon is trimmed and the remaining read is mapped to the BW25113 reference genome using bwa (accession CP009273.1) to generate insertion plot files, which we

viewed in Artemis [130]. These data can be viewed online at our browser: http://tradis-vault.qfab.org/. The plot files are input to the tradis_gene_insert_sites script to calculate the number of insertions per gene (insertion index score). Unless otherwise stated, 5% trim was applied to both the 5′ and 3′ end of each gene. The tradis_essentiality.R script was used to calculate the probability of belonging to each mode representing essential and non-essential gene populations respectively. Genes that do not meet the statistical threshold for a binary essential classification are classed as "ambiguous". The tradis_comparison.R script was used to compare read depth between control and condition samples, per gene, using a threshold of >2-fold change and a Q-value< 0.01. Plot files generated by the BioTraDIS analysis were also used as inputs for AlbaTraDIS. AlbaTraDIS (version 1.0.1) was used for comparative analysis between suppressor TIS datasets [81]. The reference genome annotation was used to define gene boundaries. We set the minimum log counts per million threshold at 10, for all other settings the default conditions were used.

## Whole genome sequencing

Whole genome sequencing was done by MicrobesNG, University of Birmingham, UK, or GENEWIZ, China, using Illumina platforms generating short read data. Sequencing data were aligned to the *E. coli* BW25113 reference genome available from the NCBI database (CP009273.1) using bwa mem and then converted to bam files and sorted and indexed using SAMtools [131,132]. Data are available at the European Nucleotide Archive (accession: PRJEB43420). The programs Snippy and VarScan were used to identify SNPs and indels, BreSeq was used to identify large chromosomal rearrangement events [133–135].

## Phylogenetic analysis

All searches were performed against the UniProt database of Reference Proteomes (v2020_06, 02-Dec-2020), which includes only complete proteomes of reference organisms, to confidently determine presence as well as absence. Search was performed with PF06295.13 using hmmsearch (3.1b2; [136]) with the—max setting. Search results were analysed manually as well as by all-against-all blast and subsequent clustering in cytoscape, to determine the search cut-off used. Taxonomy information was equally obtained from the UniProt Reference Proteomes ftp site (v2020_06, 02-Dec-2020). Phylogenetic inference was performed on the sequences aligned via mafft [137], and tree calculation using IQ-TREE [138] with the built-in model test [139], which resulted in LG+G4. Tree visualisation was performed using iTol [140].

## Supporting information

**S1 Table. Polymyxin B TIS screen data.**
(XLSX)

**S2 Table. Transposon library construction metrics.**
(XLSX)

**S3 Table. Insertion index scores of the *yhcB* and wild-type transposon libraries.**
(XLSX)

**S4 Table. Gene fitness in a *yhcB* mutant strain.**
(XLSX)

**S5 Table. PANTHER analysis.**
(XLSX)

**S6 Table. AlbaTraDIS identification of suppressor mutations.**
(XLSX)

**S7 Table. Mutations identified in spontaneous revertant suppressor mutants.**
(XLSX)

**S8 Table. Mass spectrometry data of TLC spot.**
(TXT)

**S9 Table. Peptidoglycan composition.**
(XLSX)

**S10 Table. FlaGs gene identifiers.**
(TXT)

**S11 Table. Strains used in this study.**
(DOCX)

**S12 Table. Plasmids used in this study.**
(DOCX)

**S13 Table. Oligonucleotides used in this study.**
(DOCX)

**S1 Fig. Screening the BW25113 transposon library in sub-inhibitory concentrations of polymyxin B.** (A) 0.2 μg/ml of polymyxin B in 50 ml LB does not inhibit growth of *E. coli* BW25113, under the conditions used to screen the library. One representative growth curve is shown, consistent with 5 repeats. (B) Calculation of the number of mapped reads needed to sample a given percentage of the transposon library. The equation $I = s - s\left(\frac{s-1}{s}\right)^{n}$ was used to estimate the number of mapped reads required to sample a given proportion of the dataset. s = 1,000,000 was taken as the total number of possible mutants (total transposon insertion sites). I = insertions identified, n = number of mapped reads. This data was used to calculate the approximate percentage of unique insertions identified for a given number of mapped reads, for a library of 1,000,000 unique mutants. (C) Comparison of insertion index scores (a measure of insertion density) per gene for each replicate of the library either exposed to LB only, or LB supplemented with polymyxin B. The scatter plots show the correlation coefficient for insertion density of each gene between replicates. (D) The relative abundance of Cluster of Orthologous Groups (COG) categories for all genes of BW25113 (grey) and for the 54 genes identified as required for growth in sub-inhibitory concentrations of polymyxin B (PxB; blue). (E) The transposon insertion profiles of *yqjA* and *ydbH* following outgrowth in LB only (grey, above) or in LB supplemented with polymyxin B (blue, below). The transposon insertion position along each gene is marked by a vertical line. The vertical line size corresponds with read depth, with visibly fewer transposon-insertion sites identified within *ydbH* and *yqjA* following outgrowth in polymyxin B.
(TIF)

**S2 Fig. Complementation of a ΔyhcB defect.** (A) 10-fold serial-dilution of overnight cultures grown in LB, inoculated onto LB agar plates supplemented with 0.2% arabinose, with and without 1% SDS + 0.5 mM EDTA. Strains are carrying a pBad-Myc-His-A with or without the *yhcB* CDS under the control of an arabinose promoter. (B) DIC microscopy images of day cultures grown in LB supplemented with 0.4% arabinose and 100 μg/ml carbenicillin (to maintain plasmids), with a 5 μm scale bar for reference.
(TIF)

**S3 Fig. The effect of different growth media on phenotype.** 10-fold serial-dilution of overnight cultures of the parent strain *E. coli* BW25113 (BW) and Δ*yhcB* strain (Y) grown to late stationary phase under different conditions (LB at 37˚C; M9 + 0.4% glucose at 37˚C; LB + 0.4% glucose at 37˚C; M9 + 0.4% glycerol at 37˚C; LB at 16˚C) normalized to an $OD_{600}$ of 1.00 and inoculated onto LB agar plates supplemented with and without 1% SDS; 0.1 mM EDTA or 50 μg/ml vancomycin.
(TIF)

**S4 Fig. Domains and conserved motifs of YhcB.** (A) PSIPRED secondary structure prediction of YhcB. (B) Conserved residues of YhcB predicted by ConSurf. Conserved 'NPF' and 'PRDY' motifs are highlighted with amber boxes in both panels.
(TIF)

**S5 Fig. Conservation of the PRDY domain.** Phylogenetic analysis displaying conservation of YhcB in bacterial reference genomes. Branches of the tree, and second outermost track, are coloured according to taxonomic Order. The outermost track is coloured according to the amino acid residues within the 'PXDY' domain conserved among species. The label 'other' represents those that had a different sequence to the four listed.
(TIF)

**S6 Fig. Construction of a transposon library in BW25113 parent strain.** (A) Comparison of insertion index scores (IIS) between technical replicates of each library. (B) The BW25113 transposon library. A genome map of BW25113 starting at the annotation origin, with the sense and antisense coding sequences of BW25113 shown in blue, respectively, and the position and frequency of sequenced transposon insertion events shown in grey. (C) The transposon insertion profile of *ftsH* shown for both libraries. Peaks represent the abundance of detected transposon insertion events for each library: wild-type (grey, above) and *yhcB* (pink, below) with read frequency capped at 10. (D) PCR amplification of the *yhcB* and *ftsH* loci to confirm *yhcB*-deletion and *ftsH*-replacement with a kanamycin resistance cassette in these strains.
(TIF)

**S7 Fig. Biosynthetic pathways required in a Δ*yhcB* background.** Transposon insertion data of the *yhcB* library shown in pink, below, control library shown in grey, above, the gene track with all insertion data capped at a frequency of 1. (A) Transposon insertion profile of the *tol-pal* operon. (B) The heptose biosynthetic pathway, required for LPS core biosynthesis. (C) Schematic of the ECA biosynthesis pathway adapted from Jorgenson *et al.* [80]. (D) Representative LB agar plates of colonies recovered following P1 transduction, representative of n = 2 experiments. Abbreviations: Enterobacterial Common Antigen (ECA); Undecaprenyl phosphate (Und-P); Phosphate (P); Adenosine diphosphate (ADP); deoxythymidine diphosphate (dTDP); Uridine diphosphate (UDP).
(TIF)

**S8 Fig. Transposon insertion sites within *nlpI* and *fabF* that restore resistance of a *yhcB* mutant to vancomycin or SDS and EDTA.** (A and B) Transposon insertion data for suppressor screens. Red and blue vertical lines indicate the transposon insertion position and correspond with the transposon orientation at the point of insertion. The height of the bar corresponds with mapped sequencing read frequency. The top track represents data for the library plated on LB. The tracks underneath represent the Δ*yhcB* transposon library plated on LB supplemented with SDS and EDTA, or vancomycin, at lethal doses to the BW25113Δ*yhcB* parent strain. The red boxes underneath each suppressor dataset represent significant

differential abundance of insertions in the condition sample compared to the control, identified by AlbaTraDIS. (C) Overnight cultures of the wild-type and Δ*yhcB* strain grown alongside isogenic single and double gene-deletion mutants constructed by P1-transduction using the Keio library strains as donors. Cells were grown in LB at 37˚C, normalised to an $OD_{600}$ of 1.00 and 10-fold serially diluted before inoculating LB agar plates supplemented with SDS and EDTA or vancomycin.
(TIF)

**S9 Fig. Natural suppressor mutations that are restorative in a *yhcB* mutant.** (A) Validation of natural suppressor mutants. 10-fold serial dilutions of *E. coli* K-12 BW25113, BW25113Δ*yhcB* and six BW25113Δ*yhcB* revertant suppressor mutants, grown on LB, LB supplemented with 200 μg/ml vancomycin, or 1% SDS + 0.5 mM EDTA. (B) PCR amplification of the *yhcB* locus to confirm *yhcB*-deletion in these strains. B = BW25113; Y = BW25113Δ*yhcB*; 1–6 = BW25113Δ*yhcB* suppressor mutants 1–6. (C) DIC images of BW25113, BW25113Δ*yhcB* and six BW25113Δ*yhcB* revertant suppressor mutants grown overnight in LB. Scale bar of 5 μm. The precise genotype for each mutant is listed in S7 Table: mut 1 has a single nucleotide polymorphism in *lpxC*; mut 2 has a genomic inversion resulting in an *mla* null phenotype; mut 3 and 5 have single nucleotide polymorphisms in *cdsA*; mut 4 and 6 have mutations consistent with the previously described *mla\**. (D) Solved structure of *Thermatoga maritima* CdsA (blue) and the predicted structure of *E. coli* CdsA (orange), with the surface electrostatic potential of each shown below. The deep red and deep blue colours indicate electronegative and electropositive regions at −10 and 10 kT $e^{-1}$, respectively. An overlay of the cation-binding pocket of CdsA from *T. maritima* and *E. coli* is shown in the bottom right panel. The equivalent mutated residues are annotated: shown in black, above for *T. maritima* and red, below for *E. coli*. Note the S223C residue mutated in *T. maritima* to achieve a resolved structure.
(TIF)

**S10 Fig. The *yhcB* mutant contains extra membrane and has an altered cell envelope.** (A) *E. coli* K-12 BW25113 and BW25112Δ*yhcB* grown in LB, labelled with FM4-64FX and Mito-Tracker Green FM. Images were taken at 3 h and are representative of n = 3 experiments. Lipid accumulation is indicated by the white arrows. Scale bar of 5 μm. (B-C) Thin layer chromatography (TLC) separation of phospholipid species in chloroform:methanol:water (65:25:4) solvent system, and stained with PMA. (B) Samples grown in LB and collected at two stages of growth. (C) Samples grown overnight in LB or M9-glucose. (D) TLC/autoradiographic analysis of [$^{32}$P]-labeled lipid A extracted from mid-logarithmic growth and overnight growth cultures of BW25113 and Δ*yhcB* strains grown in LB. As a positive control for lipid A palmitoylation, BW25113 cells were treated with 25 mM EDTA for 10 min prior to extraction. Abbreviations: Phosphatidylethanolamine (PE); Phosphatidylglycerol (PG); Cardiolipin (CL).
(TIF)

**S11 Fig. TEM images of BW25113 and the Δ*yhcB* mutant.** Transmission electron micrographs of (A) BW25113 and (B) the Δ*yhcB* mutant cells (standard fixation and processing). Scale bar = 1 μm in the top six images. A large Δ*yhcB* mutant cell is shown with a scale bar of 5 μm and two regions of excess or ruffled membrane structures are highlighted by red boxes (annotated in Fig 10A); enlarged images of these sections are shown underneath with a scale bar of 500 nm each.
(TIF)

**S12 Fig. Images of a Δ*yhcB* mutant cells.** Electron micrographs of fast frozen/freeze-substituted *yhcB* mutant cells. Internal membranes indicated by red arrowheads; stacked membrane arrays indicated by yellow arrowheads. Bars are: (A) 2 μm; (B) 2 μm; (C) 500 nm; (D) 200 nm;

(E) 500 nm. Note that panels C-E (ii) are higher magnification views from C-E (i), respectively.
(TIF)

**S13 Fig. Mass spectrometry analysis of the unknown lipid species identified by TLC.** (A) The chemical structure of lyso-phosphatidylethanolamine (LPE) 16:1. (B) Product ion ms/ms spectra of the three most abundant LPE species detected by mass spectrometry. The spectra show characteristic loss of 141 Da from the molecular ion $[M+H]^+$, corresponding to the loss of the PE lipid head group.
(TIF)

**S14 Fig. Membrane lipid composition analysis.** (A) A representative [$^3$H]-distribution profile of cell lysate fractions separated by centrifugation across a sucrose gradient, taken from cell cultures grown overnight in LB. (B) Ratio of [$^{32}$P]-phosphate labeled phospholipids (PL) between the outer membrane (OM) and inner membrane (IM), and between OM LPS and PLs, for both exponential and stationary growth phase cell cultures, in triplicate. Error bars represent standard deviations. Student's *t* tests: N.S., not significant, as compared to wild type (WT).
(TIF)

**S15 Fig. Analysis of the peptidoglycan composition.** Representative HPLC chromatograms showing the muropeptide composition of BW25113 and BW25113Δ*yhcB* at exponential and stationary phase. Purified peptidoglycan was digested with cellosyl and the resulting muropeptides were reduced with sodium borohydride and separated by HPLC. The major muropeptides (No. 1–17) are quantified in S9 Table.
(TIF)

**S16 Fig. UppS and CdsA conservation.** The number, and overlap, of bacterial genomes containing a UppS (K00806) or CdsA (K00981) homolog obtained from AnnoTree (v1.2.0) using the KEGG identifiers and the following search criteria: % identity: 30; E value: 0.00001; % subject alignment: 70; % query alignment: 70. (B) Tree representation of the phylogeny of bacterial genomes with species containing both UppS and CdsA highlighted in blue.
(TIF)

**S17 Fig. *uppS* and *cdsA* neighborhood conservation across the bacterial kingdom.** The conservation of the *uppS* gene neighbourhood represented by 62 diverse bacterial species. *uppS* is depicted in black, and *cdsA* in green (1). The remaining gene identifiers can be found in S10 Table. Data calculated using FlaGs [109].
(TIF)

**S18 Fig. The effect of *cdsA* knockdown on cell morphology.** DIC images collected on the Inverted LSM 880 Fast Airyscan (63x/1.40 OIL) for stationary phase cultures grown overnight in LB at 37˚C without induction. Leaky expression of the antisense *cdsA* transcript from the T7 promoter was sufficient to affect changes in morphology. Scale bar = 10 μm in column A, scale bar = 5 μm in columns B-D.
(TIF)

## Acknowledgments

We thank Stephen Trent for the generous gift of LpxC antibodies. We thank Makrina Totsika and Yaoqin Hong for the pTrc99A-*lpxC* expression vector, in addition to your time and support over the project. Thank you Yaoqin Hong for all your helpful discussion and insight into

cell envelope biology. We thank Eugenio Sanchez-Moran at the University of Birmingham for use of his microscope, and we thank the University of Queensland Microscopy department at the Institute of Molecular Biosciences for assistance in setting up the lipid-labelling microscopy experiments. In particular, we thank Nicholas Condon for your time and teaching. The authors acknowledge the use of the Microscopy Australia Research Facility at the Centre for Microscopy and Microanalysis at The University of Queensland. We thank Rick Webb for expert help with cryoEM processing. Light microscopy was performed at the Australian Cancer Research Foundation (ACRF)/Institute for Molecular Bioscience Cancer Biology Imaging Facility, which was established with the support of the ACRF.We thank MicrobesNG for providing a fantastic service generating WGS data, especially Emily Richardson for your patience and sharing your expertise.

## Author Contributions

**Conceptualization:** Emily C. A. Goodall, Jack A. Bryant, Ian R. Henderson.

**Data curation:** Emily C. A. Goodall, Georgia L. Isom, Christopher Icke, Gabriela Boelter, Alun Jones, Eva Heinz, Manuel Banzhaf, Waldemar Vollmer, Jack A. Bryant, Ian R. Henderson.

**Formal analysis:** Emily C. A. Goodall, Georgia L. Isom, Jessica L. Rooke, Karthik Pullela, Christopher Icke, Zihao Yang, Gabriela Boelter, Alun Jones, Rochelle Da Costa, Bing Zhang, James Rae, Wee Boon Tan, Matthias Winkle, Antoine Delhaye, Eva Heinz, Adam F. Cunningham, Robert G. Parton, Jeff A. Cole, Manuel Banzhaf, Shu-Sin Chng, Waldemar Vollmer, Jack A. Bryant, Ian R. Henderson.

**Funding acquisition:** Jean-Francois Collet, Adam F. Cunningham, Mark A. Blaskovich, Robert G. Parton, Ian R. Henderson.

**Investigation:** Emily C. A. Goodall, Georgia L. Isom, Jessica L. Rooke, Karthik Pullela, Christopher Icke, Zihao Yang, Gabriela Boelter, Alun Jones, Isabel Warner, Rochelle Da Costa, Bing Zhang, James Rae, Wee Boon Tan, Matthias Winkle, Antoine Delhaye, Eva Heinz, Waldemar Vollmer, Jack A. Bryant, Ian R. Henderson.

**Methodology:** Emily C. A. Goodall, Georgia L. Isom, Jessica L. Rooke, Karthik Pullela, Christopher Icke, Zihao Yang, Gabriela Boelter, Alun Jones, Isabel Warner, Rochelle Da Costa, Bing Zhang, James Rae, Wee Boon Tan, Matthias Winkle, Antoine Delhaye, Eva Heinz, Mark A. Blaskovich, Robert G. Parton, Shu-Sin Chng, Waldemar Vollmer, Jack A. Bryant, Ian R. Henderson.

**Project administration:** Mark A. Blaskovich, Robert G. Parton, Jeff A. Cole, Ian R. Henderson.

**Resources:** Jessica L. Rooke, Jean-Francois Collet, Mark A. Blaskovich, Ian R. Henderson.

**Supervision:** Emily C. A. Goodall, Jessica L. Rooke, Karthik Pullela, Christopher Icke, Jean-Francois Collet, Adam F. Cunningham, Mark A. Blaskovich, Robert G. Parton, Jeff A. Cole, Manuel Banzhaf, Shu-Sin Chng, Jack A. Bryant, Ian R. Henderson.

**Validation:** Emily C. A. Goodall, Georgia L. Isom, Jessica L. Rooke, Karthik Pullela, Christopher Icke, Zihao Yang, Gabriela Boelter, Alun Jones, Isabel Warner, Rochelle Da Costa, Wee Boon Tan, Matthias Winkle, Manuel Banzhaf, Shu-Sin Chng, Waldemar Vollmer, Jack A. Bryant.

**Visualization:** Emily C. A. Goodall, Jessica L. Rooke, Karthik Pullela, Christopher Icke, James Rae, Wee Boon Tan, Matthias Winkle, Antoine Delhaye, Eva Heinz, Robert G. Parton, Manuel Banzhaf, Shu-Sin Chng, Waldemar Vollmer.

**Writing – original draft:** Emily C. A. Goodall, Ian R. Henderson.

**Writing – review & editing:** Emily C. A. Goodall, Georgia L. Isom, Jessica L. Rooke, Karthik Pullela, Christopher Icke, Gabriela Boelter, Alun Jones, Isabel Warner, Rochelle Da Costa, Bing Zhang, James Rae, Wee Boon Tan, Matthias Winkle, Antoine Delhaye, Eva Heinz, Jean-Francois Collet, Adam F. Cunningham, Mark A. Blaskovich, Robert G. Parton, Jeff A. Cole, Manuel Banzhaf, Shu-Sin Chng, Waldemar Vollmer, Jack A. Bryant, Ian R. Henderson.

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
