## [Decision Letter · Decision Letter 0]

17 Jun 2021

Dear Dr henderson,

Thank you very much for submitting your Research Article entitled 'YhcB coordinates peptidoglycan and LPS biogenesis with phospholipid synthesis during Escherichia coli cell growth.' to PLOS Genetics.

The manuscript was fully evaluated at the editorial level and by two independent peer reviewers. The reviewers appreciated the attention to an important problem, but raised some substantial concerns about the current manuscript. Based on the reviews, we will not be able to accept this version of the manuscript, but we would be willing to review a much-revised version. We cannot, of course, promise publication at that time. In particular, it is essential that you add additional experiments to address main points #2, #3, #4 and #5 of reviewer 1 as well as #2 and #6 of reviewer 2. Also, the two reviewers make several suggestions to edit the manuscript for clarity.

If you decide to revise the manuscript for further consideration at PLOS Genetics, please aim to resubmit within the next 60 days, unless it will take extra time to address the concerns of the reviewers, in which case we would appreciate an expected resubmission date by email to plosgenetics@plos.org.

[LINK]

We are sorry that we cannot be more positive about your manuscript at this stage. Please do not hesitate to contact us if you have any concerns or questions.

Yours sincerely,

Lotte Søgaard-Andersen

Section Editor: Prokaryotic Genetics

PLOS Genetics

Reviewer's Responses to Questions

**Comments to the Authors:**

Reviewer #1: Goodall et al. set out to identify E. coli mutants with defective cell envelope by conducting a genome-wide screen for increased sensitivity to polymyxin B using TraDIS. After identifying yhcB in this screen, the authors focused on characterizing a yhcB deletion mutant through a combination of phenotypic analyses. They also conducted genome-wide screens to identify transposon insertions that have either a positive or a negative synthetic interaction with an yhcB deletion allele. Based on their findings, they concluded that YhcB coordinates peptidoglycan and LPS biogenesis with phospholipid synthesis.

Investigating how cells coordinate the biogenesis of various envelope layers is very important to understanding bacterial growth and devising new antibacterial strategies. Although previous studies have shown that YhcB is required for proper envelope biogenesis, the data presented here advances our understanding by suggesting that YhcB is important for proper balance of lipid synthesis. Nevertheless, how YhcB functions and even which of the phenotypes reported here and elsewhere are directly (or indirectly) the result of the loss of YhcB remain unclear.

The strongest part of this manuscript is the work presented in Figs. 5-10. Results from the synthetic genetic interactions and electron microscopy led the authors to conclude that YhcB is important for balanced lipid synthesis. The weakest aspects of the manuscript include the repetition of phenotypes that have already been reported before (defects in envelope permeability, shape, and survival), overstating some claims as facts instead of suggestions or models, and lacking some key data regarding YhcB’s effect on lipid synthesis.

Specific main comments:

1) In my opinion, the manuscript should focus on the novel data presented here and the role of YhcB on lipid synthesis. Figures 2-4 could be supplementary or deleted given that they have been reported by others.

2) Genome-wide screens like the ones reported here which search for mutations that display either a positive or a negative synthetic effect in an yhcB mutant are powerful, but they should be complemented with more detailed analyses of the genetic interactions revealed by the screens. This manuscript fails to do this. As a result, it is difficult to understand the extent of the negative and positive interactions reported in this work, despite the strong conclusions stated by the authors. For example:

a) Are the combinations deemed synthetic lethal in Figs. 6 and 7 truly lethal (as the authors state) or do the double mutants grow significantly worse than the single mutants?

b) Is an ftsH null allele truly not lethal in a yhcB mutant unlike in the wild type? The incidence of transposons in ftsH is still quite low in the yhcB mutant. Given that there are known suppressors of the essentiality of ftsH, the low incidence of transposition might indicate that the ftsH yhcB mutant has acquired additional mutations that allow its survival.

c) How strong is the suppression reported by the loss of Mla? Are the phenotypes (TEM and lipid changes) reported in Fig 10 for the yhcB mutant corrected by mla mutations?

There is a lot of data included in the manuscript because of the nature of the screens, but the depth of knowledge gained from this study is limited because the authors do not analyze mutants in depth. The exception is the yhcB single mutant, but a lot of that data has already been published (as the authors cite, although do not always make clear).

3) The synthetic genetic interactions suggest that yhcB mutants have an imbalance in the synthesis of total levels of phospholipids, LPS, and undecaprenyl phosphate, but the authors do not address if this is the case. Are the total levels of LPS and phospholipids changed in the yhcB mutant? If affected, they could also test to see if suppressors fix the imbalance.

4) Are suppressors of sensitivity in an yhcB mutant specific to the loss of YhcB or are they mutations that always increase resistance? The authors use the suppressor data to make their case for a role of YhcB in coordinating lipid synthesis. However, given that the suppressor screen was for increased resistance, their argument would be strengthened if suppressors were shown to be specific to the loss of YhcB. For example, it was previously reported (but not noted in the manuscript) that cdsA mutations confer a general increase in vancomycin resistance, even in a wild-type cell (PMID: 24957626).

5) Discussion lines 757-591: There is no data demonstrating that they have decoupled these phenotypes because the authors only measured growth at one concentration for each treatment. Sensitivity could be decreased somewhat but not enough to make a difference in growth in the specific concentration used by the authors. If they are going to make this claim, the authors should assess sensitivity using more quantitative methods (like minimal inhibitory concentrations). This also applies to the data from Fig 4 and the conclusions regarding the PRDY domain of YhcB only being required for resistance to some compounds. Otherwise, what’s the explanation for the difference?

Minor comments:

1) Line 28: Change “kingdoms” to “domains”.

2) Line 31: change “most well” to “best”.

3) Line 34: Change "whole genome phenotypic" to "whole-genome phenotypic".

4) Line 38: The authors state that they report “the complete” genetic interaction network of yhcB. It might be complete for transposon insertion mutants, but that is all. There might be other members of the network that could not be found because of the use of transposon mutagenesis. Remove “the complete”.

5) Line 42: I suggest changing "sits” to " functions".

6) Lines 51-52: Doesn't the cell envelope also include the periplasm?

7) Lines 62-77: The references used to cite the machineries that are involved in cell envelope biogenesis, elongation, and division need to be revised. Several of the cited reviews are from 5-10 years ago, so newer ones should be considered. In addition, the referencing of the primary literature is incomplete. The authors cited papers reporting the discovery of some but not all the components of the various machines.

8) Lines 92-93: The authors state that they demonstrate that loss of YhcB affects cell width and length but do not reference that this was shown previously by others. The images here are similar to those shown by Sung et al. However, Mehla et al reported increased length but reduced diameters. Can the authors comment on this difference?

9) Line 95: delete "all". How do you know that you found all genes that suppress the envelope defects caused by the loss of yhcB? Possibly transposon mutant alleles, but not necessarily all genes.

10) Line 97: I suggest changing "is stationed" to "functions".

11) Fig. 2A: the strains label for the mutant is missing and the wild-type strain's label is misplaced.

12) Lines 154-156: It appears from the images on Fig S2A that over-production of YhcB is somewhat toxic in wild-type and mutant strains. I suggest removing any description about length and instead comment on the detrimental effect on growth. The effect in length is not very convincing based on the data shown.

13) Lines 186-188: Do they lose viability or is it that because they filament they are expected to yield a lower number of colonies?

14) Line 218: remove “all”.

15) Line 301: What is meant by LPS stability?

16) Fig. 9 suggests that there is a preferred directionality in transposon insertion based on the color of the transposon (red is predominant, whatever direction that is since information is not provided; likely internal promoter drives expression of the locus). Is there a preference in directionality? If there is, the authors should comment on it. Later, there is data on uppS and cdsA that should have been mentioned here as well.

17) Lines 374-392: The nature of the mla* mutation should be explained in more detail to readers. Also, May et al. was not the original paper reporting the mla* allele, the Sutterlin et al 2016 PNAS paper was.

18) Line 381: what does a variation of mlaA* mutation mean? Be specific.

19) MlaA* works in the opposite way as wild-type MlaA. How do the authors explain that both mlaA* and mlaA nulls suppress?

20) Line 438: Fig. S10 should be S12.

21) Line 486: Fig. S10D should be S12D. In addition, this figure needs to include a control showing what happens when the wbbL plasmid is introduced into the wild-type strain to make sure that their conclusion is warranted. Are the colonies growing suppressors ?

22) Lines 502-504: I suggest deleting the sentence “With … resolution”.

23) Lines 506-508: Rephrase this sentence as it is confusing. The data support that the defect is most severe in LB than in minimal, but not during rapid growth in rich media. As the authors state in their next sentence, defects are most severe during stationary phase in rich media.

24) Lines 513-517: I am confused by the reference to Harris and Theriot since the yhcB mutant has both increased diameter and length, so the volume is greatly affected and not maintained.

25) Fig. 11 and legend: References showing interactions should be cited in legend. What is the difference between dotted and solid arrows?

26) Line 533-535: The images presented by the authors showing the effect in size (Fig, 2) do not resemble those shown in Vadia et al 2017. The yhcB mutants appear to have a septation defect (as shown by others), while those in Vadia did not.

27) Line 557: “Given the location of YhcB”… What do the authors mean by this? The reported co-localization with the divisome reported by Mehla et al?

28) Line 593: remove the reference to “all of the cell envelope biogenesis pathways”. There is no evidence that YhcB affects all pathways.

29) Methods: More detailed explanation of plasmid constructions should be given.

Reviewer #2: This comprehensive study provides new insight into cell envelope homeostasis and particularly the role played by YhcB. Prior work reported interaction between YhcB and the PG biosynthesis elongasome and the current data places YhcB into a broader physiological circuitry. These are interesting and original findings, connect to previous observations, and offer a broader context but a definitive mechanism is not yet available. There are some areas where changes to the presentation or additional information are warranted.

Main comments:

1) Some of the early data on envelope defects in the delta-yhcB mutant is hard to follow as currently described.

a) Fig 2A. Please label the mutant sample; I assume this is the right dilution column in each plate panel.

b) There is a clear difference in LB versus M9-Glc in Fig 2A to the extent that SDS/EDTA arguably has little effect in cells grown in M9 Glc. The way the data is described in lines 139-147 fits the LB phenotype but not M9-Glc. Some (but not all) of this is clarified later but it was frustrating to try and sort it out while working through the results sequentially

c) The text implies the complementation data is in Fig 2A but it isn’t. Fig. S2A needs to be referenced in the sentence that begins on line 144 “Ectopic expression….”.

d) Fig S2A (right) shows an effect of transforming the WT with pBAD-yhcB that goes away with Ara induction. Please explain this observation.

e) How do the authors explain 2 different cell morphology phenotypes for the mutant grown in M9? Is this due to an accumulation of additional mutations in the population?

2) Dissection of YhcB functional domains.

a) What is EV in Fig 4C – I presume it is empty vector?

b) Line 205-206 – the text suggests Fig 4A offers data on the functional importance of the PRDY data; it does not, it only shows conservation.

c) The data points to the critical part of YhcB being cytoplasmic. How does this connect to the locations of other proteins identified by the screen, including “components of the elongasome” (line 264).

3) RodZ essentiality.

a) Please explain how insertions in the 5-prime end of the gene are not polar on expression of the 3-prime essential region.

b) please describe the topology of RodZ to explain how it can interact with the cytoplasmic part of YhcB; this might not be obvious to a non-specialist.

b) the Discussion (line 555) states that loss of the elongasome is restorative. Do the authors mean complete loss of elongasome activity or substantial reduction in elongasome activity?

4) Suppressors of yhcB mutant defects.

a) The observation that insertions upstream of uppS and cdsA are suppressors is very interesting. To confirm the later interpretations, it would be helpful to provide some additional confirmatory insight i.e. do these mutants lead to altered transcription of the locus?

b) Some additional data relevant to ECA and UppS appears in the last section of the results and I think it would be clearer if this were unified with the earlier discussion. Otherwise it seems a bit repetitive.

5) The membrane ruffling visualized by simple TEM in Fig 10A and Fig S10 does not seem so convincing, considering the severe phenotypes. TEM does not seem the right approach here. The freeze substitution images in Figs 10B S11 and S12 are far more compelling. I suggest removing the TEM image from Fig 10 and deleting Fig S10 since these are mostly redundant.

6) Accumulation of LPE in the yhcB mutant. This seems an important observation and I believe it merits a more robust identification than that provided by TLC migration and correlated PagP activity. This should be amenable to MS given the abundance of the TLC spot.

The text (line 440) refers to Fig 10B for LPE identification; it should be 10C.

Other editorial points:

Where YhcB is first mentioned in the Introduction, it would be helpful to include the few references specific to this protein (e.g. doi: 10.1016/j.jbc.2021.100700; 10.1007/s12275-020-0078-4; 10.5402/2012/304021).

The initial screen/selection was done with 0.2 micrograms/mL; what is the MIC for this strain?

Line 301: I am not sure “LPS stability” is the right term here. The read-out is OM stability dictated by LPS structure.

Line 313: this references relevant work from Young’s group on sequestration of undecaprenyl phosphate in ECA defects. A recent paper from Troutman’s group should also be added (doi: 10.1021/acschembio.0c00983).

Line 466: the reference to Fig S11 is incorrect.

How many replicates were done for the PG analysis?

Line 575-577: How does conservation of uppS-cdsA reveal a tight linkage between PL production and polysaccharide biosynthesis.

**Have all data underlying the figures and results presented in the manuscript been provided?**

Reviewer #1: Yes

Reviewer #2: Yes

PLOS authors have the option to publish the peer review history of their article (what does this mean?). If published, this will include your full peer review and any attached files.

Reviewer #1: No

Reviewer #2: No

---

## [Decision Letter · Decision Letter 1]

27 Oct 2021

Dear Dr henderson,

Thank you very much for submitting your Research Article entitled 'YhcB coordinates peptidoglycan and LPS biogenesis with phospholipid synthesis during Escherichia coli cell growth.' to PLOS Genetics.

The manuscript was fully evaluated at the editorial level and by independent peer reviewers. The reviewers appreciated the attention to an important topic but identified some concerns that we ask you address in a revised manuscript. In particular, I encourage you to carefully consider the comments made by reviewer #1 and modify your manuscript accordingly. At this point, we do not request any additional experiments.

We therefore ask you to modify the manuscript according to the review recommendations. Your revisions should address the specific points made by each reviewer.

[LINK]

Yours sincerely,

Lotte Søgaard-Andersen

Section Editor: Prokaryotic Genetics

PLOS Genetics

Lotte Søgaard-Andersen

Section Editor: Prokaryotic Genetics

PLOS Genetics

Reviewer's Responses to Questions

**Comments to the Authors:**

Reviewer #1: The authors have significantly improved the manuscript by adding new data that strengthens their findings and conclusions. I appreciate their efforts and improvements, especially those involving the reconstruction of the key mutants and lipid analyses.

There are still some issues:

1) The main conclusion, as highlighted in the title, abstract, and throughout the manuscript is that YhcB coordinates peptidoglycan and LPS biogenesis with phospholipid synthesis. In some cases, YhcB is referred to as regulator (e. g. lines 57-59, 651). The authors present evidence connecting YhcB to these processes through phenotypes and/or genetic connections. However, they do not provide evidence that YhcB coordinates these processes. As the authors state, growing cells coordinate peptidoglycan, LPS, and phospholipid biogenesis. Therefore, a factor that affects all three processes could be acting as a true coordinating factor of two or more of these processes or be a factor that directly affects only one of them and indirectly affects the other two because of the inherent coordination that exists in the cell. In fact, the authors state (lines 550-553) that the “loss of yhcB results in an increased demand for the Und-PP synthase, resulting in upregulation of the uppS-cdsA operon. Increased expression of cdsA in turn gives rise to excess phospholipid synthesis, resulting in detergent sensitivity and enlarged cells.” In other words, the authors explain that the phenotypes they observe in the yhcB mutant result from an unexplained deficiency of Und-PP that ends up upregulating phospholipid synthesis simply because a gene for phospholipid synthesis (cdsA) is in an operon with uppS, which encodes for a Und-PP synthase. Furthermore, the connections to LPS biogenesis appears to come from genetic interactions that ultimately decrease phospholipid biosynthesis by increasing LPS synthesis through their balanced synthesis. This study provides a lot on information on YhcB but we still do not really know its function and what the primary defect is resulting from its loss. The authors provide a reasonable explanation supporting an indirect effect on phospholipid synthesis, which ultimately also links it to LPS biogenesis. Therefore, the authors need to change their title and assertions that YhcB coordinates peptidoglycan, LPS, and phospholipid biogenesis. The running title should also be edited.

2) Readers will have to commit a significant amount of time to read and go over the data presented in this manuscript. Given the type and amount of data in the paper, I think that it would be extremely helpful to readers if key data from some of the supplementary tables that the authors highlight in the figures and text is presented in an easily accessible form. This is specifically relevant to Tables S3, S4, and S6. For example, the excel file for Table S3 could have an additional sheet displaying the list of the 28 genes highlighted in the text and Fig 4B as being essential in the wild type and not in the yhcB mutant. The same approach could be used to report the list of genes contributing to fitness reported in Table S4. Otherwise, readers need to sort through the information, and some readers might not have the knowledge to do it. In fact, I spent quite a bit of time doing this, and my numbers do not match those reported by the authors. It is possible that I made mistakes or was not given all the information I needed, reinforcing the importance of providing a concise list of the relevant data (in addition to the complete data), especially to understand why the authors focused on select data. After sorting the table S3 using the authors’ annotations of "essential" and "non-essential", I only found 16 genes, not 28, as essential in the wild type but not the mutant, and 108, instead of 121, as essential in the yhcB strain but not the wild type. The numbers I obtained from Table S4 do not match either; likely, I obtained larger numbers because the authors filtered out essential genes when reporting the relevant data and I did not because that information is not provided in Table S4. For Table S6 (and text in line 365), it is unclear which of the suppressors listed in Table S6 are the 28 "universal suppressors” since cross-referencing both lists yields 43 in common (40 genes and 3 intergenic regions).

3) On the analysis of the different domains of YhcB: The authors never tested the mutant that only lacks the helical domain, so it's a bit of a stretch to state that the “data suggest a dual function of the YhcB protein: the cytoplasmic helical domain is needed to suppress sensitivity to SDS and EDTA, while the function of the ‘PRDY’ domain is critical for survival in the presence of vancomycin, but not polymyxin B or SDS and EDTA” (lines 234-237). More importantly, these data do not necessarily indicate a dual function. At the very least, having two functions is not the only explanation for their results. What the authors are reporting could result from a gradient of defects in one single function. It seems that some mutants can grow on SDS/EDTA plates but not on vancomycin plates. Also, slow growth restores growth on SDS/EDTA but not vancomycin. One possible explanation for all these observations is that vancomycin plates, at the concentration used, demand more of the function that is defective in yhcB mutants than SDS/EDTA plates at the concentration used. In fact, the authors state that to "confirm the barrier defect of the yhcB mutant, we grew the parent strain, E. coli BW25113, and our isogenic DyhcB mutant on LB agar plates supplemented with either vancomycin, a high molecular weight antibiotic that is typically unable to cross the Gram-negative OM, or plates supplemented with SDS and EDTA" (lines 163-166). Since sensitivity to vancomycin, SDS/EDTA, and polymyxin result from a "barrier defect", the statement in lines 163-166 seems to be at odds with using SDS/EDTA and vancomycin to discriminate between different functions. If the authors want to propose two functions for YhcB based on differential sensitivity of the domain mutants, they need to explain their reasoning in a way that does not disagree with their explanations for how their chemical selections and phenotypic analyses work.

4) The authors stated that the mla* allele is toxic in the wild type but not in the yhcB mutant. I assume this refers to the stationary-phase death reported for the mla* mutant by others. The authors showed that mlaA* suppresses yhcB, but I do not think they showed that the stationary-phase death in the mla* mutant is corrected in the yhcB mutant. It might be true, but it is not known unless I missed it. These data are not required, the authors can just remove their claim (might be in a couple of places).

5) The authors did not remove the sentence "With thousands of deleterious and restorative genetic interactions discovered, to the best of our knowledge no other gene has been scrutinised genetically at such scale and resolution" (lines 557-559) because they “believe it to be true”. It is disappointing and quite telling that they feel the need to keep an unnecessary statement that can only be substantiated with their “belief”. Moreover, "thousands" of genetic interactions must imply that they are counting every transposon insertion as one interaction. So, if a gene has 100 different insertions and all of them simply result in functionally equivalent null alleles, the authors are counting them as 100 genetic interactions. Although technically correct, it is an interesting choice of words to describe their work. As a reviewer, I appreciate that the authors chose to write the manuscript at the gene level, not the insertion level.

6) The authors’ attempt to fix the sentence regarding counting and enumerating the parts on the cell envelope is still problematic. The revised sentence reads: “In Gram-negative bacteria, the tripartite cell envelope is composed of the cytoplasmic membrane (CM), the peptidoglycan (PG) layer within the periplasmic space and an outer membrane (OM)” (lines 66-68). This sentence is still incorrect. “Tripartite” indicates 3 parts but there are 4: CM, PG layer, periplasmic space, and OM. Based on the original and revised sentences, it appears that the authors do not recognize the periplasm as a part of the cell envelope. This is quite surprising and makes me wonder if they consider the cytoplasm a part of the cell.

7) In response to the authors’ answer about the question of whether suppressors are specific to the loss of yhcB (response to point 4 by reviewer 1): Their answer provided is partly confusing. Their argument about the mla suppressors makes sense since the single mutants are sensitive but the double mutants are more resistant to SDS/EDTA. However, the rest of the answer is irrelevant to the point of the question. It remains to be known how specific other suppressors are. I'm not asking for the authors to test them but to acknowledge this issue in their text. I hope the authors also recognize that the validation of "some of their findings" does not give a blanket validation of all other mutants as they imply in their answer.

8) Authors’ response to Reviewer 1, point 9: although the change is acceptable, I want to clarify the reason for my previous request since the authors were not sure. Your study used transposon mutagenesis to generate mutants and reports on a few spontaneous suppressors. The sentence was an absolute statement about having found "all" genes that suppress the yhcB null. The old sentence was incorrect because 1) only the transposon mutant collection was saturated in this study and there are other types of alleles that might suppress that could never be generated by transposons, and 2) there is no indication that the spontaneous selections were saturated.

9) Lines 56-57: “YhcB influences the temporal biogenesis of LPS, peptidoglycan and membrane phospholipids.” I am not sure why the authors specify a “temporal” effect as opposed to, for example, an effect on the balance of their synthesis (amount). If the temporal reference relates to an excess phospholipid synthesis occurring in stationary phase, then it would be more appropriate to refer to the physiological state of the cell (not time) but would raise questions about YhcB’s role during exponential growth.

10) Lines 159-160: “We observed that the �yhcB mutants from the Keio library do not phenocopy each other.” The new statement is confusing. Is it that there is more than one yhcB mutant in the collection, or that different colonies from one single mutant behaved differently, or that the same mutant from different copies of the collection in different labs do not behave the same? In addition, in the Methods section, it seems that the authors constructed a new yhcB mutant by moving the allele from their keio collection into their wild type by transduction based on their description. Given that they raise concerns about the mutant in their Keio collection, it might be useful to readers if they clarified that if allele in the Keio collection is correct and can be used if moved into a new strain.

11) Line 364: given that mutants were generated by transposition, referring to gene-deletion suppressor mutations is incorrect.

12) Line 590: Rephrase "mutations within the metal ion binding site of CdsA" since mutations occur in DNA, and the relevant amino acids are located near but not within the metal-binding site.

13) Lines 595 and 602: Please specify (to clarify to readers) that the listed findings pertain to the yhcB mutant.

14) Line 614: “it’s” should be “its”

15) Table S1 and lines 148-149: yjbS was not in category "S" even though is defined as "uncharacterized protein" in Table S1?

16) Table S7: in the excel file, it is mislabeled as Table S8. Also, what is a “conservative in-frame” deletion/insertion? One that conserves the reading frame? Isn’t that denoted by “in-frame”?

17) It would also be helpful if the authors kindly helped readers by adding the relevant genotypes to the labels in Fig. S9 instead of only using “mut#” labels.

Reviewer #2: The authors have given careful consideration to my comments and I am satisfied with the changes and response to comments. One minor point the authors might consider is to briefly comment on the crystal structure in the "domains of YhcB..." section to offer more context.

**Have all data underlying the figures and results presented in the manuscript been provided?**

Reviewer #1: Yes

Reviewer #2: Yes

PLOS authors have the option to publish the peer review history of their article (what does this mean?). If published, this will include your full peer review and any attached files.

Reviewer #1: No

Reviewer #2: No

---

## [Editor Report · Decision Letter 2]

24 Nov 2021

Dear Dr henderson,

We are pleased to inform you that your manuscript entitled "Loss of YhcB results in dysregulation of coordinated peptidoglycan, LPS and phospholipid synthesis during Escherichia coli cell growth" has been editorially accepted for publication in PLOS Genetics. Congratulations!

Yours sincerely,

Lotte Søgaard-Andersen

Section Editor: Prokaryotic Genetics

PLOS Genetics

Lotte Søgaard-Andersen

Section Editor: Prokaryotic Genetics

PLOS Genetics

Comments from the reviewers (if applicable):

**Data Deposition**

http://datadryad.org/submit?journalID=pgenetics&manu=PGENETICS-D-21-00648R2

**Press Queries**

---

## [Editor Report · Acceptance letter]

20 Dec 2021

PGENETICS-D-21-00648R2 

Loss of YhcB results in dysregulation of coordinated peptidoglycan, LPS and phospholipid synthesis during *Escherichia coli* cell growth 

Dear Dr henderson, 

We are pleased to inform you that your manuscript entitled "Loss of YhcB results in dysregulation of coordinated peptidoglycan, LPS and phospholipid synthesis during *Escherichia coli* cell growth" has been formally accepted for publication in PLOS Genetics! Your manuscript is now with our production department and you will be notified of the publication date in due course.

With kind regards,

Zsofia Freund

PLOS Genetics

On behalf of:
